# Causal Inference through a Witness Protection Program

**Ricardo Silva**
Department of Statistical Science and CSML
University College London
ricardo@stats.ucl.ac.uk

**Robin Evans**
Department of Statistics
University of Oxford
evans@stats.ox.ac.uk

## Abstract

One of the most fundamental problems in causal inference is the estimation of a causal effect when variables are confounded. This is difficult in an observational study because one has no direct evidence that all confounders have been adjusted for. We introduce a novel approach for estimating causal effects that exploits observational conditional independencies to suggest "weak" paths in a unknown causal graph. The widely used faithfulness condition of Spirtes et al. is relaxed to allow for varying degrees of "path cancellations" that will imply conditional independencies but do not rule out the existence of confounding causal paths. The outcome is a posterior distribution over bounds on the average causal effect via a linear programming approach and Bayesian inference. We claim this approach should be used in regular practice to complement other default tools in observational studies.

## 1   Contribution

We provide a new methodology to bound the average causal effect (ACE) of a variable $X$ on a variable $Y$. For binary variables, the ACE is defined as

$$E[Y \,|\, do(X = 1)] - E[Y \,|\, do(X = 0)] = P(Y = 1 \,|\, do(X = 1)) - P(Y = 1 \,|\, do(X = 0)), \quad (1)$$

where $do(\cdot)$ is the operator of Pearl [14], denoting distributions where a set of variables has been intervened upon by an external agent. In the interest of space, we assume the reader is familiar with the concept of causal graphs, the basics of the $do$ operator, and the basics of causal discovery algorithms such as the PC algorithm of Spirtes et al. [22]. We provide a short summary for context in Section 2

The ACE is in general not identifiable from observational data. We obtain upper and lower bounds on the ACE by exploiting a set of (binary) covariates, which we also assume are not effects of $X$ or $Y$ (justified by temporal ordering or other background assumptions). Such covariate sets are often found in real-world problems, and form the basis of most observational studies done in practice [21]. However, it is not obvious how to obtain the ACE as a function of the covariates. Our contribution modifies the results of Entner et al. [6], who exploit conditional independence constraints to obtain point estimates of the ACE, but give point estimates relying on assumptions that might be unstable in practice. Our modification provides a different interpretation of their search procedure, which we use to generate candidate *instrumental variables* [11]. The linear programming approach of Dawid [5] and Ramsahai [16] is then modified to generate bounds on the ACE by introducing constraints on some causal paths, motivated as relaxations of [6]. The new setup can be computationally expensive, so we introduce further relaxations to the linear program to generate novel symbolic bounds, and a fast algorithm that sidesteps the full linear programming optimization with some simple, message passing-like, steps.

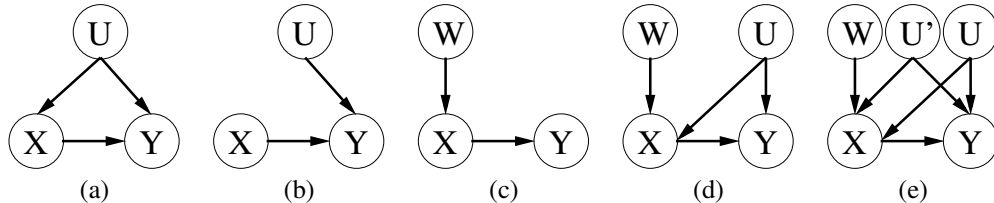

Figure 1: (a) A generic causal graph where $X$ and $Y$ are confounded by some $U$. (b) The same system in (a) where $X$ is intervened upon by an external agent. (c) A system where $W$ and $Y$ are independent given $X$. (d) A system where it is possible to use faithfulness to discover that $U$ is sufficient to block all back-door paths between $X$ and $Y$. (e) Here, $U$ itself is not sufficient.

Section 2 introduces the background of the problem and Section 3 our methodology. Section 4 discusses an analytical approximation of the main results, and a way by which this provides scaling-up possibilities for the approach. Section 5 contains experiments with synthetic and real data.

## 2 Background: Instrumental Variables, Witnesses and Admissible Sets

Assuming $X$ is a potential cause of $Y$, but not the opposite, a cartoon of the causal system containing $X$ and $Y$ is shown in Figure 1(a). $U$ represents the universe of common causes of $X$ and $Y$. In control and policy-making problems, we would like to know what happens to the system when the distribution of $X$ is overridden by some external agent (e.g., a doctor, a robot or an economist). The resulting modified system is depicted in Figure 1(b), and represents the family of distributions indexed by $do(X = x)$: the graph in (a) has undergone a "surgery" that wipes out edges, as originally discussed by [22] in the context of graphical models. Notice that if $U$ is observed in the dataset, then we can obtain the distribution $P(Y = y \mid do(X = x))$ by simply calculating $\sum_u P(Y = y \mid X = x, U = u)P(U = u)$ [22]. This was popularized by [14] as the *back-door adjustment*. In general $P(Y = y \mid do(X = x))$ can be vastly different from $P(Y = y \mid X = x)$.

The ACE is simple to estimate in a randomized trial: this follows from estimating the conditional distribution of $Y$ given $X$ under data generated as in Figure 1(b). In contrast, in an *observational study* [21] we obtain data generated by the system in Figure 1(a). If one believes all relevant confounders $U$ have been recorded in the data then back-door adjustment can be used, though such completeness is uncommon. By postulating knowledge of the causal graph relating components of $U$, one can infer whether a measured subset of the causes of $X$ and $Y$ is enough [14, 23, 15]. Without knowledge of the causal graph, assumptions such as *faithfulness* [22] are used to infer it.

The faithfulness assumption states that a conditional independence constraint in the observed distribution exists if and only if a corresponding structural independence exists in the underlying causal graph. For instance, observing the independence $W \perp\!\!\!\perp Y \mid X$, and assuming faithfulness and the causal order, we can infer the causal graph Figure 1(c); in all the other graphs this conditional independence in not implied. We deduce that no unmeasured confounders between $X$ and $Y$ exist. This simple procedure for identifying chains $W \to X \to Y$ is useful in exploratory data analysis [4], where a large number of possible causal relations $X \to Y$ are unquantified but can be screened using observational data before experiments are performed. The idea of using faithfulness is to be able to sometimes identify such quantities.

Entner et al. [6] generalize the discovery of chain models to situations where a non-empty set of covariates is necessary to block all back-doors. Suppose $\mathcal{W}$ is a set of covariates which are known not to be effects of either $X$ or $Y$, and we want to find an *admissible set* contained in $\mathcal{W}$: a set of observed variables which we can use for back-door adjustment to get $P(Y = y \mid do(X = x))$. Entner's "Rule 1" states the following:

**Rule 1***: If there exists a variable $W \in \mathcal{W}$ and a set $\mathbf{Z} \subseteq \mathcal{W}\backslash\{W\}$ such that:*

$$(i) \quad W \not\perp\!\!\!\perp Y \mid \mathbf{Z} \qquad\qquad (ii) \quad W \perp\!\!\!\perp Y \mid \mathbf{Z} \cup \{X\}.$$

*then infer that $\mathbf{Z}$ is an admissible set.*

A point estimate of the ACE can then be found using $\mathbf{Z}$. Given that $(W, \mathbf{Z})$ satisfies[1] Rule 1, we call $W$ a *witness* for the admissible set $\mathbf{Z}$. The model in Figure 1(c) can be identified with Rule 1, where $W$ is the witness and $\mathbf{Z} = \emptyset$. In this case, a so-called Naïve Estimator[2] $P(Y = 1 \,|\, X = 1) - P(Y = 1 \,|\, X = 0)$ will provide the correct ACE. If $U$ is observable in Figure 1(d), then it can be identified as an admissible set for witness $W$. Notice that in Figure 1(a), taking $U$ as a scalar, it is not possible to find a witness since there are no remaining variables. Also, if in Figure 1(e) our covariate set $\mathcal{W}$ is $\{W, U\}$, then no witness can be found since $U'$ cannot be blocked. Hence, it is possible for a procedure based on Rule 1 to answer "I don't know whether an admissible set exists" even when a back-door adjustment would be possible *if* one knew the causal graph. However, using the faithfulness assumption alone one cannot do better: Rule 1 is complete for non-zero effects without more information [6].

Despite its appeal, the faithfulness assumption is not without difficulties. Even if unfaithful distributions can be ruled out as pathological under seemingly reasonable conditions [13], distributions which lie close to (but not on) a simpler model may in practice be indistinguishable from distributions within that simpler model at finite sample sizes. To appreciate these complications, consider the structure in Figure 1(d) with $U$ unobservable. Here $W$ is randomized but $X$ is not, and we would like to know the ACE of $X$ on $Y$[3]. $W$ is sometimes known as an *instrumental variable* (IV), and we call Figure 1(d) the *standard IV structure*; if this structure is known, optimal bounds $\mathcal{L}_{IV} \leq$ ACE $\leq \mathcal{U}_{IV}$ can be obtained without further assumptions, using only observational data over the binary variables $W$, $X$ and $Y$ [1]. There exist distributions faithful to the IV structure but which at finite sample sizes may appear to satisfy the Markov property for the structure $W \rightarrow X \rightarrow Y$; in practice this can occur at any finite sample size [20]. The true average causal effect may lie anywhere in the interval $[\mathcal{L}_{IV}, \mathcal{U}_{IV}]$ (which can be rather wide), and may differ considerably from the naïve estimate appropriate for the simpler structure. While we emphasize that this is a 'worst-case scenario' analysis and by itself should not rule out faithfulness as a useful assumption, it is desirable to provide a method that gives greater control over violations of faithfulness.

## 3   Methodology: the Witness Protection Program

The core of our idea is (i) to *invert the usage of Entner's Rule 1*, so that pairs $(W, \mathbf{Z})$ should provide an instrumental variable bounding method instead of a back-door adjustment; (ii) express violations of faithfulness as *bounded violations of local independence*; (iii) find bounds on the ACE using *a linear programming formulation*.

Let $(W, \mathbf{Z})$ be any pair found by a search procedure that decides when Rule 1 holds. $W$ will play the role of an instrumental variable, instead of being discarded. A standard IV bounding procedure such as [1] can be used conditional on each individual value $\mathbf{z}$ of $\mathbf{Z}$, then averaged over $P(\mathbf{Z})$. The lack of an edge $W \rightarrow Y$ given $\mathbf{Z}$ can be justified by faithfulness (as $W \perp\!\!\!\perp Y \,|\, \{X, \mathbf{Z}\}$). For the same reason, there might be no (conditional) dependence between $W$ and a possible unmeasured common parent of $X$ and $Y$. However, assuming faithfulness itself is not interesting, as a back-door adjustment could be directly obtained. Allowing unconstrained dependencies induced by edges $W \rightarrow Y$ and $(W, U)$ (any direction) is also a non-starter, as all bounds will be vacuous [16].

Consider instead the (partial) parameterization in Table 1 of the joint distribution of $\{W, X, Y, U\}$, where $U$ is latent and not necessarily a scalar. For simplicity of presentation, assume we are conditioning everywhere on a particular value $\mathbf{z}$ of $\mathbf{Z}$, but which we supress from our notation as this will not be crucial to developments in this Section. Under this notation, the ACE is given by

$$\eta_{11}P(W = 1) + \eta_{10}P(W = 0) - \eta_{01}P(W = 1) - \eta_{00}P(W = 0). \tag{2}$$

$$
\begin{aligned}
\zeta^{\star}_{yx.w} &\equiv P(Y=y, X=x \,|\, W=w, U) \\
\zeta_{yx.w} &\equiv \textstyle\sum_U P(Y=y, X=x \,|\, W=w, U)P(U \,|\, W=w) \\
&= P(Y=y, X=x \,|\, W=w) \\[4pt]
\eta^{\star}_{xw} &\equiv P(Y=1 \,|\, X=x, W=w, U) \\
\eta_{xw} &\equiv \textstyle\sum_U P(Y=1 \,|\, X=x, W=w, U)P(U \,|\, W=w) \\
&= P(Y=1 \,|\, do(X=x), W=w) \\[4pt]
\delta^{\star}_{w} &\equiv P(X=1 \,|\, W=w, U) \\
\delta_{w} &\equiv \textstyle\sum_U P(X=x \,|\, W=w, U)P(U \,|\, W=w) \\
&= P(X=1 \,|\, W=w).
\end{aligned}
$$

Table 1: A partial parameterization of a causal DAG model over some $\{U, W, X, Y\}$. Notice that such parameters cannot be functionally independent, and this is precisely what we will exploit.

We now introduce the following assumptions,

$$|\eta^{\star}_{x1} - \eta^{\star}_{x0}| \le \epsilon_w \tag{3}$$

$$|\eta^{\star}_{xw} - P(Y=1 \,|\, X=x, W=w)| \le \epsilon_y \tag{4}$$

$$|\delta^{\star}_{w} - P(X=1 \,|\, W=w)| \le \epsilon_x \tag{5}$$

$$\underline{\beta}P(U) \le P(U \,|\, W=w) \le \bar{\beta}P(U). \tag{6}$$

Setting $\epsilon_w = 0$, $\underline{\beta} = \bar{\beta} = 1$ recovers the standard IV structure. Further assuming $\epsilon_y = \epsilon_x = 0$ recovers the chain structure $W \to X \to Y$. Deviation from these values corresponds to a violation of faithfulness, as the premises of Rule 1 can only be satisfied by enforcing functional relationships among the conditional probability tables of each vertex. Using this parameterization in the case $\epsilon_y = \epsilon_x = 1$, $\underline{\beta} = \bar{\beta} = 1$, Ramsahai [16], extending [5], used the following linear programming to obtain bounds on the ACE (for now, assume that $\zeta_{yx.w}$ and $P(W=w)$ are known constants):

1. There is a 4-dimensional polytope where parameters $\{\eta^{\star}_{xw}\}$ can take values: for $\epsilon_w = \epsilon_y = 1$, this is the unit hypercube $[0,1]^4$. Find the extreme points of this polytope (up to 12 points for the case where $\epsilon_w > 0$). Do the same for $\{\delta^{\star}_{w}\}$.

2. Find the extreme points of the joint space $\zeta^{\star}_{yx.w}$ by mapping them from the points in $\{\delta^{\star}_{w}\} \times \{\eta^{\star}_{xw}\}$, since $\zeta^{\star}_{yx.w} = (\delta^{\star}_{w})^x(1 - \delta^{\star}_{w})^{(1-x)}\eta^{\star}_{xw}$.

3. Using the extreme points of the 12-dimensional joint space $\{\zeta^{\star}_{yx.w}\} \times \{\eta^{\star}_{xw}\}$, find the dual polytope of this space in terms of linear inequalities. Points in this polytope are convex combinations of $\{\zeta^{\star}_{yx.w}\} \times \{\eta^{\star}_{xw}\}$, shown by [5] to correspond to the marginalization over some arbitrary $P(U)$. This results in contraints over $\{\zeta_{yx.w}\} \times \{\eta_{xw}\}$.

4. Maximize/minimize (2) with respect to $\{\eta_{xw}\}$ subject to the constraints found in Step 3 to obtain upper/lower bounds on the ACE.

Allowing for the case where $\epsilon_x < 1$ or $\epsilon_y < 1$ is just a matter changing the first step, where box constraints are set on each individual parameter as a function of the known $P(Y=y, X=x \,|\, W=w)$, prior to the mapping in Step 2. The resulting constraints are now implicitly non-linear in $P(Y=y, X=x \,|\, W=w)$, but at this stage this does not matter as they are treated as constants. To allow for the case $\underline{\beta} < 1 < \bar{\beta}$, use exactly the same procedure, but substitute every occurrence of $\zeta_{yx.w}$ in the constraints by $\kappa_{yx.w} \equiv \sum_U \zeta^{\star}_{yx.w}P(U)$; notice the difference between $\kappa_{yx.w}$ and $\zeta_{yx.w}$. Likewise, substitute every occurrence of $\eta_{xw}$ in the constraints by $\omega_{xw} \equiv \sum_U \eta^{\star}_{xw}P(U)$. Instead of plugging in constants for the values of $\kappa_{yx.w}$ and turning the crank of a linear programming solver, we first treat $\{\kappa_{yx.w}\}$ (and $\{\omega_{xw}\}$) as unknowns, linking them to observables and $\eta_{xw}$ by the constraints $\zeta_{yx.w}/\bar{\beta} \le \kappa_{yx.w} \le \zeta_{yx.w}/\underline{\beta}$, $\sum_{yx} \kappa_{yx.w} = 1$ and $\eta_{xw}/\bar{\beta} \le \omega_{xw} \le \eta_{xw}/\underline{\beta}$. Finally, the method can be easily implemented using a package such as Polymake (http://www.poymake.org) or SCDD for R. More details are given in the Supplemental Material.

In this paper, we will not discuss in detail how to choose the free parameters of the relaxation. Any choice of $\epsilon_w \ge 0, \epsilon_y \ge 0, \epsilon_x \ge 0, 0 \le \underline{\beta} \le 1 \le \bar{\beta}$ is *guaranteed to provide bounds that are at*

**input** : Binary data matrix $\mathcal{D}$; set of relaxation parameters $\theta$; covariate index set $\mathcal{W}$;
cause-effect indices $X$ and $Y$

**output**: A list of pairs (witness, admissible set) contained in $\mathcal{W}$

$\mathcal{L} \leftarrow \emptyset$;
**for** *each* $W \in \mathcal{W}$ **do**
    **for** *every admissible set* $\mathbf{Z} \subseteq \mathcal{W}\backslash\{W\}$ *identified by* $W$ *and* $\theta$ *given* $\mathcal{D}$ **do**
        $\mathcal{B} \leftarrow$ posterior over upper/lowed bounds on the ACE as given by $(W, \mathbf{Z}, X, Y, \mathcal{D}, \theta)$;
        **if** *there is no evidence in* $\mathcal{B}$ *to falsify the* $(W, \mathbf{Z}, \theta)$ *model* **then**
            $\mathcal{L} \leftarrow \mathcal{L} \cup \{\mathcal{B}\}$;
        **end**
    **end**
**end**
**return** $\mathcal{L}$

**Algorithm 1:** The outline of the Witness Protection Program algorithm.

*least as conservative* as the back-door adjusted point estimator of [6], which is always covered by the bounds. Background knowledge, after a user is suggested a witness and admissible set, can be used here. In Section 5 we experiment with a few choices of default parameters. To keep focus, in what follows we will discuss only computational aspects. We develop a framework for choosing relaxation parameters in the Supplemental, and expect to extend it in follow-up publications.

As the approach provides the witness a degree of protection against faithfulness violations, using a linear program, we call this framework the *Witness Protection Program* (WPP).

## 3.1 Bayesian Learning

The previous section treated $\zeta_{yx.w}$ and $P(W = w)$ as known. A common practice is to replace them by plug-in estimators (and in the case of a non-empty admissible set $\mathbf{Z}$, an estimate of $P(\mathbf{Z})$ is also necessary). Such models can also be falsified, as the constraints generated are typically only supported by a strict subset of the probability simplex. In principle, one could fit parameters without constraints, and test the model by a direct check of satisfiability of the inequalities using the plug-in values. However, this does not take into account the uncertainty in the estimation. For the standard IV model, [17] discuss a proper way of testing such models in a frequentist sense.

Our models can be considerably more complicated. Recall that constraints will depend on the extreme points of the $\{\zeta_{yx.w}^\star\}$ parameters. As implied by (4) and (5), extreme points will be functions of $\zeta_{yx.w}$. Writing the constraints fully in terms of the observed distribution will reveal non-linear relationships. We approach the problem in a Bayesian way. We will assume first the dimensionality of $\mathbf{Z}$ is modest (say, 10 or less), as this is the case in most applications of faithfulness to causal discovery. We parameterize $P(Y, X, W \mid \mathbf{Z})$ as a full $2 \times 2 \times 2$ contingency table[4].

Given that the dimensionality of the problem is modest, we assign to each three-variate distribution $P(Y, X, W \mid \mathbf{Z} = \mathbf{z})$ an independent Dirichet prior for every possible assigment of $\mathbf{Z}$, constrained by the inequalities implied by the corresponding polytopes. The posterior is also a 8-dimensional constrained Dirichlet distribution, where we use rejection sampling to obtain a posterior sample by proposing from the unconstrained Dirichlet. A Dirichlet prior can also be assigned to $P(\mathbf{Z})$. Using a sample from the posterior of $P(\mathbf{Z})$ and a sample (for each possible value $\mathbf{z}$) from the posterior of $P(Y, X, W \mid \mathbf{Z} = \mathbf{z})$, we obtain a sample upper and lower bound for the ACE.

The full algorithm is shown in Algorithm 1. The search procedure is left unspecified, as different existing approaches can be plugged in into this step. See [6] for a discussion. In Section 5 we deal with small dimensional problems only, using the brute-force approach of performing an exhaustive search for $\mathbf{Z}$. In practice, brute-force can be still valuable by using a method such as discrete PCA [3] to reduce $\mathcal{W}\backslash\{W\}$ to a small set of binary variables. To decide whether the premises in Rule 1 hold, we merely perform Bayesian model selection with the BDeu score [2] between the full graph $\{W \rightarrow X, W \rightarrow Y, X \rightarrow Y\}$ (conditional on $\mathbf{Z}$) and the graph with the edge $W \rightarrow Y$ removed. Our

$$\omega_{xw} \geq \kappa_{1x.w} + L_{xw}^{YU}(\kappa_{0x'.w} + \kappa_{1x'.w}) \tag{7}$$

$$\omega_{xw} \leq 1 - (\kappa_{0x.w'} - \epsilon_w(\kappa_{0x.w'} + \kappa_{1x.w'}))/U_{xw'}^{XU} \tag{8}$$

$$\omega_{xw} - \omega_{xw'}U_{x'w}^{XU} \leq \kappa_{1x.w} + \epsilon_w(\kappa_{0x'.w} + \kappa_{1x'.w}) \tag{9}$$

$$\omega_{xw} + \omega_{x'w} - \omega_{x'w'} \geq \kappa_{1x'.w} + \kappa_{1x.w} - \kappa_{1x'.w'} + \kappa_{1x.w'} - \chi_{xw'}(\bar{U} + \underline{L} + 2\epsilon_w) + \underline{L} \tag{10}$$

Table 2: Some of the algebraic bounds found by symbolic manipulation of linear inequalities. Notation: $x, w \in \{0, 1\}$, $x' = 1 - x$ and $w' = 1 - w$ are the complementary values. $L_{xw}^{YU} \equiv \max(0, P(Y = 1 | X = x, W = w) - \epsilon_y)$, $U_{xw}^{YU} \equiv \min(1, P(Y = 1 | X = x, W = w) + \epsilon_y)$; $L_{xw}^{XU} \equiv \max(0, P(X = x | W = w) - \epsilon_x)$, with $U_{xw}^{XU}$ defined accordingly. Finally, $\bar{U} \equiv \max\{U_{xw}^{YU}\}$, $\underline{L} \equiv \min\{L_{xw}^{YU}\}$ and $\chi_{xw} \equiv \kappa_{1x.w} + \kappa_{0x.w}$. Full set of bounds with proofs can be found in the Supplementary Material.

"falsification test" in Step 5 is a simple and pragmatical one: our initial trial of rejection sampling proposes $M$ samples, and if more than 95% of them are rejected, we take this as an indication that the proposed model provides a bad fit. The final result is a set of posterior distributions over bounds, possibly contradictory, which should be summarized as appropriate. Section 5 provides an example.

## 4 Algebraic Bounds and the Back-substitution Algorithm

Posterior sampling is expensive within the context of Bayesian WPP: constructing the dual polytope for possibly millions of instantiations of the problem is time consuming, even if each problem is small. Moreover, the numerical procedure described in Section 3 does not provide any insight on how the different free parameters $\{\epsilon_w, \epsilon_y, \epsilon_x, \beta, \bar{\beta}\}$ interact to produce bounds, unlike the analytical bounds available in the standard IV case. [16] derives analytical bounds under (3) given a *fixed*, *numerical* value of $\epsilon_w$. We know of no previous analytical bounds as an algebraic function of $\epsilon_w$.

In the Supplementary Material, we provide a series of algebraic bounds as a function of our free parameters. Due to limited space, we show only some of the bounds in Table 2. They illustrate qualitative aspects of our free parameters. For instance, if $\epsilon_y = 1$ and $\beta = \bar{\beta} = 1$, then $L_{xw}^{YU} = 0$ and (7) collapses to $\eta_{xw} \geq \zeta_{1x.w}$, one of the original relations found by [1] for the standard IV model. Decreasing $\epsilon_y$ will linearly increase $L_{xw}^{YU}$, tightening the corresponding lower bound in (7). If also $\epsilon_w = 0$ and $\epsilon_x = 1$, from (8) it follows $\eta_{xw} \leq 1 - \zeta_{0x.w'}$. Equation (3) implies $\omega_{x'w} - \omega_{x'w'} \leq \epsilon_w$, and as such by setting $\epsilon_w = 0$ we have that (10) implies $\eta_{xw} \geq \eta_{1x.w} + \eta_{1x.w'} - \eta_{1x'.w'} - \eta_{0x.w'}$, one of the most complex relationships in [1]. Further geometric intuition about the structure of the binary standard IV model is given by [19].

These bounds are not tight, in the sense that we opted not to fully exploit all possible algebraic combinations for some results, such as (10): there we use $\underline{L} \leq \eta_{xw}^{\star} \leq \bar{U}$ and $0 \leq \delta_w^{\star} \leq 1$ instead of all possible combinations resulting from (4) and (5). The proof idea in the Supplementary Material can be further refined, at the expense of clarity. Because our derivation is a further relaxation, the implied bounds are more conservative (i.e., wider).

Besides providing insight on the structure of the problem, this gives a very efficient way of checking whether a proposed parameter vector $\{\zeta_{yx.w}^{\star}\}$ is valid, as well as finding the bounds: use back-substitution on the symbolic set of constraints to find box constraints $\mathcal{L}_{xw} \leq \omega_{xw} \leq \mathcal{U}_{xw}$. The proposed parameter will be rejected whenever an upper bound is smaller than a lower bound, and (2) can be trivially optimized conditioning only on the box constraints—this is yet another relaxation, added on top of the ones used to generate the algebraic inequalities. We initialize by intersecting all algebraic box constraints (of which (7) and (8) are examples); next we refine these by scanning relations $\pm\omega_{xw} - a\omega_{xw'} \leq c$ such as (9) in lexicographical order, and tightening the bounds of $\omega_{xw}$ using the current upper and lower bounds on $\omega_{xw'}$ where possible. We then identify constraints $\mathcal{L}_{xww'} \leq \omega_{xw} - \omega_{xw'} \leq \mathcal{U}_{xww'}$ starting from $-\epsilon_w \leq \omega_{xw} - \omega_{xw'} \leq \epsilon_w$ and the existing bounds, and plug into relations $\pm\omega_{xw} + \omega_{x'w} - \omega_{x'w'} \leq c$ (as exemplified by (10)) to get refined bounds on $\omega_{xw}$ as functions of $(\mathcal{L}_{x'ww'}, \mathcal{U}_{x'ww'})$. We iterate this until convergence, which is guaranteed since bounds never widen at any iteration. This back-substitution of inequalities follows the spirit

of message-passing and it is an order of magnitude more efficient than the fully numerical solution, while not increasing the width of the bounds by too much. In the Supplementary Material we provide evidence for this claim. In our experiments in Section 5, the back-substitution method was used in the testing stage of WPP. After collecting posterior samples, we calculated the posterior expected value of the contingency tables and run the numerical procedure to obtain the final tight bound[5].

# 5  Experiments

We describe a set of synthetic studies, followed by one study with the influenza data discussed by [9, 18]. In the synthetic study setup, we compare our method against NE1 and NE2, two naïve point estimators defined by back-door adjustment on the whole of $\mathcal{W}$ and on the empty set, respectively. The former is widely used in practice, even when there is no causal basis for doing so [15]. The point estimator of [6], based solely on the faithfulness assumption, is also assessed.

We generate problems where conditioning on the whole set $\mathcal{W}$ is guaranteed to give incorrect estimates[6]. Here, $|\mathcal{W}| = 8$. We analyze two variations: one where it is guaranteed that at least one valid *witness × admissible set* pair exists; in the other, latent variables in the graph are common parents also of $X$ and $Y$, so no valid witness exists. We divide each variation into two subcases: in the first, "hard" subcase, parameters are chosen (by rejection sampling) so that NE1 has a bias of at least 0.1 in the population; in the second, no such selection exists, and as such our exchangeable parameter sampling scheme makes the problem relatively easy. We summarize each WPP bound by the posterior expected value of the lower and upper bounds. In general WPP returns more than one bound: we select the upper/lower bound corresponding to the $(W, \mathbf{Z})$ pair where the sum of BDeu scores for $W \not\perp\!\!\!\perp Y \mid \mathbf{Z}$ and $W \perp\!\!\!\perp Y \mid \mathbf{Z} \cup \{X\}$ is highest.

Our main evaluation metric for an estimate is the Euclidean distance (henceforth, "error") between the true ACE and the closed point in the given estimate, whether the estimate is a point or an interval. For methods that provide point estimates (NE1, NE2, and faithfulness), this means just the absolute value of the difference between the true ACE and the estimated ACE. For WPP, the error of the interval $[\mathcal{L}, \mathcal{U}]$ is zero if the true ACE lies in this interval. We report *error average* and *error tail mass at 0.1*, the latter meaning the proportion of cases where the error exceeds 0.1. The comparison is not straightforward, since the trivial interval $[-1, 1]$ will always have zero bias according to this definition. This is a trade-off, to be set according to an agreed level of information loss, measured by the width of the resulting intervals. This is discussed in the Supplemental. We run simulations at two levels of parameters: $\underline{\beta} = 0.9, \bar{\beta} = 1.1$, and the same configuration except for $\underline{\beta} = \bar{\beta} = 1$. The former gives somewhat wide intervals. As Manski emphasizes [11], this is the price for making fewer assumptions. For the cases where no witness exists, Entner's Rule 1 should theoretically report no solution. In [6], stringent thresholds for accepting the two conditions of Rule 1 are adopted. Instead we take a more relaxed approach, using a uniform prior on the hypothesis of independence, and a BDeu prior with effective sample size of 10. As such, due to the nature of our parameter randomization, almost always (typically $> 90\%$) the method will propose at least one witness. Given this theoretical failure, for the problems where no exact solution exists, we assess how sensitive the methods are given conclusions taken from "approximate independencies" instead of exact ones.

We simulate 100 datasets for each one of the four cases (hard case/easy case, with theoretical solution/without theoretical solution), 5000 points per dataset, 1000 Monte Carlo samples per decision. Results are summarized in Table 3 for the case $\epsilon_w = \epsilon_x = \epsilon_y = 0.2, \underline{\beta} = 0.9, \bar{\beta} = 1.1$. Notice

| Case ($\underline{\beta}=1, \bar{\beta}=1$) | NE1 | | NE2 | | Faith. | | WPP | | Width |
|---|---|---|---|---|---|---|---|---|---|
| Hard/Solvable | 0.12 | 1.00 | 0.02 | 0.03 | 0.05 | 0.05 | 0.01 | 0.01 | 0.24 |
| Easy/Solvable | 0.01 | 0.01 | 0.07 | 0.24 | 0.02 | 0.01 | 0.00 | 0.00 | 0.24 |
| Hard/Unsolvable | 0.16 | 1.00 | 0.20 | 0.88 | 0.19 | 0.95 | 0.07 | 0.25 | 0.24 |
| Easy/Unsolvable | 0.09 | 0.32 | 0.14 | 0.56 | 0.12 | 0.53 | 0.03 | 0.08 | 0.23 |

Table 3: Summary of the outcome of the synthetic studies. Each entry for particular method is a pair (bias average, bias tail mass at 0.1) of the respective methods, as explained in the main text. The last column is the median width of the WPP interval. In a similar experiment with $\underline{\beta} = 0.9, \bar{\beta} = 1.1$, WPP achieves nearly zero error, with interval widths around 0.50. A much more detailed table for many other cases is provided in the Supplementary Material.

that WPP is quite stable, while the other methods have strengths and weaknesses depending on the setup. For the unsolvable cases, we average over the approximately $99\%$ of cases where some solution was reported—in theory, no conditional independences hold and no solution should be reported, but WPP shows empirical robustness for the true ACE in these cases.

Our empirical study concerns the effect of influenza vaccination on a patient being hospitalized later on with chest problems. $X = 1$ means the patient got a flu shot, $Y = 1$ indicates the patient was hospitalized. A negative ACE therefore suggests a desirable vaccine. The study was originally discussed by [12]. Shots were not randomized, but doctors were randomly assigned to receive a reminder letter to encourage their patients to be inoculated, recorded as *GRP*. This suggests the standard IV model in Figure 1(d), with $W = GRP$ and $U$ unobservable. Using the bounds of [1] and observed frequencies gives an interval of $[-0.23, 0.64]$ for the ACE. WPP could *not* validate *GRP* as a witness, instead returning as the highest-scoring pair the witness *DM* (patient had history of diabetes prior to vaccination) with admissible set composed of *AGE* (dichotomized at 60 years) and *SEX*. Here, we excluded *GRP* as a possible member of an admissible set, under the assumption that it cannot be a common cause of $X$ and $Y$. Choosing $\epsilon_w = \epsilon_y = \epsilon_x = 0.2$ and $\underline{\beta} = 0.9, \bar{\beta} = 1.1$, we obtain the posterior expected interval $[-0.10, 0.17]$. This does *not* mean the vaccine is more likely to be bad (positive ACE) than good: the posterior distribution is over bounds, not over points, being completely agnostic about the distribution within the bounds. Notice that even though we allow for full dependence between all of our variables, the bounds are considerably stricter than in the standard IV model due to the weakening of hidden confounder effects postulated by observing conditional independences. Posterior plots and sensitivity analysis are included in the Supplementary Material; for further discussion see [18, 9].

## 6  Conclusion

Our model provides a novel compromise between point estimators given by the faithfulness assumptions and bounds based on instrumental variables. We believe such an approach should become a standard item in the toolbox of anyone who needs to perform an observational study. R code is available at `http://www.homepages.ucl.ac.uk/~ucgtrbd/wpp`. Unlike risky Bayesian approaches that put priors directly on the parameters of the unidentifiable latent variable model $P(Y, X, W, U \mid \mathbf{Z})$, the constrained Dirichlet prior does not suffer from massive sensitivity to the choice of hyperparameters, as discussed at length by [18] and the Supplementary Material. By focusing on bounds, WPP keeps inference more honest, providing a compromise between a method purely based on faithfulness and purely theory-driven analyses that overlook competing models suggested by independence constraints. As future work, we will look at a generalization of the procedure beyond relaxations of chain structures $W \rightarrow X \rightarrow Y$. Much of the machinery here developed, including Entner's Rules, can be adapted to the case where causal ordering is unknown: the search for "Y-structures" [10] generalizes the chain structure search to this case. Also, we will look into ways on suggesting plausible values for the relaxation parameters, already touched upon in the Supplementary Material. Finally, the techniques used to derive the symbolic bounds in Section 4 may prove useful in a more general context and complement other methods to find subsets of useful constraints such as the information theoretical approach of [8] and the graphical approach of [7].

**Acknowledgements.** We thank McDonald, Hiu and Tierney for their flu vaccine data, and the anonymous reviewers for their valuable feedback.

## Footnotes

[1]The work in [6] aims also at identifying zero effects with a "Rule 2". For simplicity we assume that the effect of interest was already identified as non-zero.

[2]Sometimes we use the word "estimator" to mean a functional of the probability distribution instead of a statistical estimator that is a function of samples of this distribution. Context should make it clear when we refer to an actual statistic or a functional.

[3]A classical example is in non-compliance: suppose $W$ is the assignment of a patient to either drug or placebo, $X$ is whether the patient actually took the medicine or not, and $Y$ is a measure of health status. The doctor controls $W$ but not $X$. This problem is discussed by [14] and [5].

[4]That is, we allow for dependence between $W$ and $Y$ given $\{X, \mathbf{Z}\}$, interpreting the decision of independence used in Rule 1 as being only an indicator of approximate independence.

[5]Sometimes, however, the expected contingency table given by the back-substitution method would fall outside the feasible region of the fully specified linear program – this is expected to happen from time to time, as the analytical bounds are looser. In such a situation, we report the bounds given by the back-substitution samples.

[6]In detail: we generate graphs where $\mathcal{W} \equiv \{Z_1, Z_2, \ldots, Z_8\}$. Four independent latent variables $L_1, \ldots, L_4$ are added as parents of each $\{Z_5, \ldots, Z_8\}$; $L_1$ is also a parent of $X$, and $L_2$ a parent of $Y$. $L_3$ and $L_4$ are each randomly assigned to be a parent of either $X$ or $Y$, but not both. $\{Z_5, \ldots, Z_8\}$ have no other parents. The graph over $Z_1, \ldots, Z_4$ is chosen by adding edges uniformly at random according to the lexicographic order. In consequence using the full set $\mathcal{W}$ for back-door adjustment is always incorrect, as at least four paths $X \leftarrow L_1 \rightarrow Z_i \leftarrow L_2 \rightarrow Y$ are active for $i = 5, 6, 7, 8$. The conditional probabilities of a vertex given its parents are generated by a logistic regression model with pairwise interactions, where parameters are sampled according to a zero mean Gaussian with standard deviation 10 / number of parents. Parameter values are truncated so that all conditional probabilities are between 0.025 and 0.975.

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
