[Supplementary Material]

# Causal Inference through a Witness Protection Program

**Ricardo Silva**                                          RICARDO@STATS.UCL.AC.UK
*Department of Statistical Science and CSML*
*University College London*
*London WC1E 6BT, UK*

**Robin Evans**                                             EVANS@STATS.OX.AC.UK
*Department of Statistics*
*University of Oxford*
*Oxford OX1 3TG, UK*

## Abstract

One of the most fundamental problems in causal inference is the estimation of a causal effect when variables are confounded. This is difficult in an observational study, because one has no direct evidence that all confounders have been adjusted for. We introduce a novel approach for estimating causal effects that exploits observational conditional independencies to suggest "weak" paths in a unknown causal graph. The widely used faithfulness condition of Spirtes et al. is relaxed to allow for varying degrees of "path cancellations" that imply conditional independencies but do not rule out the existence of confounding causal paths. The outcome is a posterior distribution over bounds on the average causal effect via a linear programming approach and Bayesian inference. We claim this approach should be used in regular practice along with other default tools in observational studies.

**Keywords:** Causal inference, instrumental variables, Bayesian inference, linear programming

## 1. Contribution

We provide a new methodology for obtaining bounds on the average causal effect (ACE) of a variable $X$ on a variable $Y$. For binary variables, the ACE is defined as

$$E[Y \mid do(X = 1)] - E[Y \mid do(X = 0)] = P(Y = 1 \mid do(X = 1)) - P(Y = 1 \mid do(X = 0)), \ (1)$$

where $do(\cdot)$ is the operator of Pearl (2000), denoting distributions where a set of variables has been intervened on by an external agent. In this paper, we assume the reader is familiar with the concept of causal graphs, the basics of the *do* operator, and the basics of causal discovery algorithms such as the PC algorithm of Spirtes et al. (2000). We provide a short summary for context in Section 2.

The ACE is in general not identifiable from observational data. We obtain upper and lower bounds on the ACE by exploiting a set of (binary) covariates, which we also assume are not affected by $X$ or $Y$ (justified by temporal ordering or other background assumptions). Such covariate sets are often found in real-world problems, and form the basis of many of the observational studies done in practice (Rosenbaum, 2002a). However, it is not obvious how to obtain the ACE as a function of the covariates. Our contribution modifies the results of Entner et al. (2013), who exploit conditional independence constraints to obtain

Figure 1: (a) A generic causal graph where $X$ and $Y$ are confounded by some $U$. (b) The same system in (a) where $X$ is intervened upon by an external agent. (c) A system where $W$ and $Y$ are independent given $X$. (d) A system where it is possible to use faithfulness to discover that $U$ is sufficient to block all back-door paths between $X$ and $Y$. (e) Here, $U$ itself is not sufficient.

point estimates of the ACE, but relying on assumptions that might be unstable with finite sample sizes. Our modification provides a different interpretation of their search procedure, which we use to generate candidate *instrumental variables* (Manski, 2007). The linear programming approach of Dawid (2003), inspired by Balke and Pearl (1997) and further refined by Ramsahai (2012), is then modified to generate bounds on the ACE by introducing constraints on some causal paths, motivated as relaxations of Entner et al. (2013). The new setup can be computationally expensive, so we introduce further relaxations to the linear program to generate novel symbolic bounds, and a fast algorithm that sidesteps the full linear programming optimization with some simple, message passing-like steps.

In Section 2, we briefly discuss the background of the problem. Section 3 contains our main methodology, while commenting on why the unidentifiability of the ACE matters even in a Bayesian context. Section 4 discusses an analytical approximation of the main results of the methodology, as well as a way by which this provides scaling-up possibilities for the approach. Our approach introduces free parameters, and Section 5 provides practical guidelines on how to choose them. Section 6 contains experiments with synthetic and real data.

## 2. Background: Instrumental Variables, Witnesses and Admissible Sets

Assuming $X$ is a potential cause of $Y$, but not the opposite, a cartoon of the possibly complex real-world causal system containing $X$ and $Y$ is shown in Figure 1(a). $U$ represents the universe of common causes of $X$ and $Y$. In control and policy-making problems, we would like to know what happens to the system when the distribution of $X$ is overridden by some external agent (e.g., a doctor, a robot or an economist). The resulting modified system is depicted in Figure 1(b), and represents the family of distributions indexed by $do(X = x)$: the graph in (a) has undergone a "surgery" that removes incoming edges to $X$. Spirtes et al. (2000) provide an account of the first graphical methods applying this idea, which are related to the overriding of structural equations proposed by Haavelmo (1943). Notice that if $U$ is observed in the dataset, then we can obtain the distribution $P(Y = y \mid do(X = x))$ by simply calculating $\sum_u P(Y = y \mid X = x, U = u)P(U = u)$ (Spirtes et al., 2000). This

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

To appreciate these complications, consider the structure in Figure 1(d) with $U$ unobservable. Here $W$ is randomized but $X$ is not, and we would like to know the ACE of $X$ on $Y$[3]. $W$ is sometimes known as an *instrumental variable* (IV), and we call Figure 1(d) the *standard IV structure* (SIV): the distinctive features here being the constraints $W \perp\!\!\!\perp U$ and $W \perp\!\!\!\perp Y \mid \{X, U\}$, statements which include latent variables. If this structure is known, optimal bounds

$$\mathcal{L}_{SIV} \leq E[Y \mid do(X = 1)] - E[Y \mid do(X = 0)] \leq \mathcal{U}_{SIV}$$

can be obtained without further assumptions, and estimated using only observational data over the binary variables $W$, $X$ and $Y$ (Balke and Pearl, 1997). However, there exist distributions faithful to the IV structure but which at finite sample sizes may appear to satisfy the Markov property for the structure $W \to X \to Y$; in practice this can occur at any finite sample size (Robins et al., 2003). The true average causal effect may lie anywhere in the interval $[\mathcal{L}_{SIV}, \mathcal{U}_{SIV}]$, which can be rather wide even when $W \perp\!\!\!\perp Y \mid X$, as shown by the following result:

**Proposition 1** *If $W \perp\!\!\!\perp Y \mid X$ and the model follows the causal structure of the standard IV graph, then $\mathcal{U}_{SIV} - \mathcal{L}_{SIV} = 1 - |P(X = 1 \mid W = 1) - P(X = 1 \mid W = 0)|$.*

All proofs in this manuscript are given in Appendix A. For a fixed joint distribution $P(W, X, Y)$, the length of such an interval cannot be further improved (Balke and Pearl, 1997). Notice that the length of the interval will depend on how strongly associated $W$ and $X$ are: $W = X$ implies $\mathcal{U}_{IV} - \mathcal{L}_{IV} = 0$ as expected, since this is the scenario of a perfect intervention. The scenario where $W \perp\!\!\!\perp X$ is analogous to not having any instrumental variable, and the length of corresponding interval is 1.

Thus, the true ACE may differ considerably from the Naïve Estimator, appropriate for the simpler structure $W \to X \to Y$ but not for the standard IV structure. While we emphasize that this is a 'worst-case scenario' analysis and by itself should not rule out faithfulness as a useful assumption, it is desirable to provide a method that gives greater control over violations of faithfulness.

## 3. Methodology: the Witness Protection Program

The core of our idea is (i) to *invert the usage of Entner et al.'s Rule 1*, so that pairs $(W, \mathbf{Z})$ should provide an instrumental variable bounding method instead of a back-door

---

3. A classical example is in non-compliance: suppose $W$ is the assignment of a patient to either drug or placebo, $X$ is whether the patient actually took the medicine or not, and $Y$ is a measure of health status. The doctor controls $W$ but not $X$. This problem is discussed by Pearl (2000) and Dawid (2003).

adjustment; (ii) express violations of faithfulness as *bounded violations of local independence*; (iii) find bounds on the ACE using *a linear programming formulation.*

Let $(W, \mathbf{Z})$ be any pair found by a search procedure that decides when Rule 1 holds. $W$ will play the role of an instrumental variable, instead of being discarded. Conditional on $\mathbf{Z}$, the lack of an edge $W \rightarrow Y$ can be justified by faithfulness (as $W \perp\!\!\!\perp Y \mid \{X, \mathbf{Z}\}$). For the same reason, there should not be any (conditional) dependence between $W$ and a possible unmeasured common parent[4] $U$ of $X$ and $Y$. Hence, $W \perp\!\!\!\perp U$ and $W \perp\!\!\!\perp Y \mid \{U, X\}$ hold given $\mathbf{Z}$. A standard IV bounding procedure such as (Balke and Pearl, 1997) can then be used conditional on each individual value $\mathbf{z}$ of $\mathbf{Z}$, then averaged over $P(\mathbf{Z})$. That is, we can independently obtain lower and upper bounds $\{\mathcal{L}(\mathbf{z}), \mathcal{U}(\mathbf{z})\}$ for each value $\mathbf{z}$, and bound the ACE by

$$\sum_{\mathbf{z}} \mathcal{L}(\mathbf{z}) P(\mathbf{Z} = \mathbf{z}) \leq E[Y \mid do(X = 1)] - E[Y \mid do(X = 0)] \leq \sum_{\mathbf{z}} \mathcal{U}(\mathbf{z}) P(\mathbf{Z} = \mathbf{z}), \quad (2)$$

since $E[Y \mid do(X = 1)] - E[Y \mid do(X = 0)] = \sum_{\mathbf{z}} (E[Y \mid do(X = 1), \mathbf{Z} = \mathbf{z}] - E[Y \mid do(X = 0), \mathbf{Z} = \mathbf{z}]) P(\mathbf{Z} = \mathbf{z})$.

Under the assumption of faithfulness and the satisfiability of Rule 1, the above interval estimator is redundant, as Rule 1 allows the direct use of the back-door adjustment using $\mathbf{Z}$. Our goal is to not enforce faithfulness, but use Rule 1 as a motivation to exclude arbitrary violations of faithfulness.

In what follows, assume $\mathbf{Z}$ is set to a particular value $\mathbf{z}$ and all references to distributions are implicitly assumed to be defined conditioned on the event $\mathbf{Z} = \mathbf{z}$. That is, for simplicity of notation, we will neither represent nor condition on $\mathbf{Z}$ explicitly. The causal ordering where $X$ and $Y$ cannot precede any other variable is also assumed, as well as the causal ordering between $X$ and $Y$.

Consider a standard parameterization of a directed acyclic graph (DAG) model, not necessarily causal, in terms of conditional probability tables (CPTs): let $\theta_{v.\mathbf{p}}^{V}$ represent $P(V = v \mid Par(V) = \mathbf{p})$ where $V \in \{W, X, Y, U\}$ denotes both a random variable and a vertex in the corresponding DAG; $Par(V)$ is the corresponding set of parents of $V$. Faithfulness violations occur when independence constraints among observables are not *structural*, but due to "path cancellations." This means that parameter values are arranged so that $W \perp\!\!\!\perp Y \mid X$ holds, but paths connecting $W$ and $U$, or $W$ and $Y$, may exist so that either $W \not\perp\!\!\!\perp U$ or $W \not\perp\!\!\!\perp Y \mid \{U, X\}$. In this situation, some combination of the following should hold true:

$$\begin{array}{rcl}
P(Y = y \mid X = x, W = w, U = u) & \neq & P(Y = y \mid X = x, U = u) \\
P(Y = y \mid X = x, W = w, U = u) & \neq & P(Y = y \mid X = x, W = w) \\
P(X = x \mid W = w, U = u) & \neq & P(X = x \mid W = w) \\
P(U = u \mid W = w) & \neq & P(U = u),
\end{array} \quad (3)$$

for some $\{w, x, y, u\}$ in the sample space of $P(W, X, Y, U)$.

For instance, if the second and third statements above are true and under the assumption of faithfulness, this implies the existence of an active path into $X$ and $Y$ via $U$, conditional

---

4. In this manuscript, we will sometimes refer to $U$ as a *set* of common parents, although we do not change our notation to bold face to reflect that.

Figure 2: A visual depiction of the family of assumptions introduced in our framework. Dashed edges correspond to conditional dependencies that are constrained according to free parameters, displayed along each corresponding edge. This is motivated by observing $W \perp\!\!\!\perp Y \mid X$.

on $W^5$, such as $X \leftarrow U \rightarrow Y$. If the first statement is true, this corresponds to an active path between $W$ and $Y$ into $Y$ that is not blocked by $\{X, U\}$. If the fourth statement is true, $U$ and $W$ are marginally dependent, with a corresponding active path. Notice some combinations are still compatible with a model where $W \perp\!\!\!\perp U$ and $W \perp\!\!\!\perp Y \mid \{U, X\}$ hold: if the second statement in (3) is false, this means $U$ cannot be a common parent of $X$ and $Y$. This family of models is observationally equivalent[6] to one where $U$ is independent of all variables.

When translating the conditions (3) into parameters $\{\theta_{v,\mathbf{p}}^V\}$, we need to define which parents each vertex has. In our CPT factorization, we define $Par(X) = \{W, U\}$ and $Par(Y) = \{W, X, U\}$; the joint distribution of $\{W, U\}$ can be factorized arbitrarily. In the next subsection, we refine the parameterization of our model by introducing redundancies: we provide a parameterization for the latent variable model $P(W, X, Y, U)$, the interventional distribution $P(W, Y, U \mid do(X))$ and the corresponding (latent-free) marginals $P(W, X, Y)$, $P(W, Y \mid do(X))$. These distributions parameters are related, and cannot differ arbitrarily. It is this fact that will allow us to bound the ACE using only $P(W, X, Y)$.

## 3.1 Encoding Faithfulness Relaxations with Linear Constraints

We define a *relaxation of faithfulness* as any set of assumptions that allows the relations in (3) to be true, but not necessarily in an *arbitrary* way: this means that while the left-hand and right-hand sides of each entry of (3) are indeed different, their difference is bounded by either the absolute difference or by ratios. Without such restrictions, (3) will only imply vacuous bounds of length 1, as discussed in our presentation of Proposition 1.

Consider the following parameterization of the distribution of $\{W, X, Y, U\}$ under the observational and interventional regimes, and their respective marginals obtained by in-

---

5. That is, a path that d-connects $X$ and $Y$ and includes $U$, conditional on $W$; it is "into" $X$ (and $Y$) because the edge linking $X$ to the path points to $X$. See Spirtes et al. (2000) and Pearl (2000) for formal definitions and more examples.

6. Meaning a family of models where $P(W, X, Y)$ satisfies the same constraints.

tegrating $U$ way[7]. Again we condition everywhere on a particular value $\mathbf{z}$ of $\mathbf{Z}$ but, for simplicity of presentation, we supress this from our notation, since it is not crucial to developments in this Section:

$$
\begin{aligned}
\zeta^{\star}_{yx.w} &\equiv P(Y = y, X = x \,|\, W = w, U) \\
\zeta_{yx.w} &\equiv \textstyle\sum_U P(Y = y, X = x \,|\, W = w, U) P(U \,|\, W = w) \\
&= P(Y = y, X = x \,|\, W = w) \\[6pt]
\eta^{\star}_{xw} &\equiv P(Y = 1 \,|\, X = x, W = w, U) \\
\eta_{xw} &\equiv \textstyle\sum_U P(Y = 1 \,|\, X = x, W = w, U) P(U \,|\, W = w) \\
&= P(Y = 1 \,|\, do(X = x), W = w) \\[6pt]
\delta^{\star}_{w} &\equiv P(X = 1 \,|\, W = w, U) \\
\delta_{w} &\equiv \textstyle\sum_U P(X = x \,|\, W = w, U) P(U \,|\, W = w) \\
&= P(X = 1 \,|\, W = w).
\end{aligned}
$$

Under this encoding, the ACE is given by

$$
\eta_{11} P(W = 1) + \eta_{10} P(W = 0) - \eta_{01} P(W = 1) - \eta_{00} P(W = 0). \tag{4}
$$

Notice that we do not explicitly parameterize the marginal of $U$, for reasons that will become clear later.

We introduce the following assumptions, as illustrated by Figure 2:

$$
|\eta^{\star}_{x1} - \eta^{\star}_{x0}| \leq \epsilon_w \tag{5}
$$

$$
|\eta^{\star}_{xw} - P(Y = 1 \,|\, X = x, W = w)| \leq \epsilon_y \tag{6}
$$

$$
|\delta^{\star}_{w} - P(X = 1 \,|\, W = w)| \leq \epsilon_x \tag{7}
$$

$$
\underline{\beta} P(U) \leq P(U \,|\, W = w) \leq \bar{\beta} P(U). \tag{8}
$$

Setting $\epsilon_w = 0$, $\underline{\beta} = \bar{\beta} = 1$ recovers the standard IV structure. Further assuming $\epsilon_y = \epsilon_x = 0$ recovers the chain structure $W \to X \to Y$. Under this parameterization in the case $\epsilon_y = \epsilon_x = 1$, $\underline{\beta} = \bar{\beta} = 1$, Ramsahai (2012), extending Dawid (2003), used linear programming to obtain bounds on the ACE. We will briefly describe the four main steps of the framework of Dawid (2003), and refer to the cited papers for more details of their implementation.

For now, assume that $\zeta_{yx.w}$ and $P(W = w)$ are known constants—that is, treat $P(W, X, Y)$ as known. This assumption will be dropped later. Dawid's formulation of a bounding procedure for the ACE is as follows.

**Step 1** *Notice that parameters $\{\eta^{\star}_{xw}\}$ take values in a 4-dimensional polytope. Find the extreme points of this polytope. Do the same for $\{\delta^{\star}_{w}\}$.*

In particular, for $\epsilon_w = \epsilon_y = 1$, the polytope of feasible values for the four dimensional vector $(\eta^{\star}_{00}, \eta^{\star}_{01}, \eta^{\star}_{10}, \eta^{\star}_{11})$ is the unit hypercube $[0, 1]^4$, a polytope with a total of 16 vertices $(0, 0, 0, 0), (0, 0, 0, 1), \ldots (1, 1, 1, 1)$. Dawid (2003) covered the case $\epsilon_w = 0$, where a two-dimensional vector $\{\eta^{\star}_{x}\}$ replaces $\{\eta^{\star}_{xw}\}$. In Ramsahai (2012), the case $0 \leq \epsilon_w < 1$ is also

---

7. Notice from the development in this Section that $U$ is not necessarily a scalar, nor discrete.

covered: some of the corners in $[0, 1]^4$ disappear and are replaced by others. The case where $\epsilon_w = \epsilon_x = \epsilon_y = 1$ is vacuous, in the sense that the consecutive steps cannot infer non-trivial constraints on the ACE.

**Step 2** *Find the extreme points of the joint space $\{\zeta^\star_{yx.w}\} \times \{\eta^\star_{xw}\}$ by mapping them from the extreme points of $\{\delta^\star_w\} \times \{\eta^\star_{xw}\}$, since $\zeta^\star_{yx.w} = (\delta^\star_w)^x(1 - \delta^\star_w)^{(1-x)}\eta^\star_{xw}$.*

The extreme points of the joint space $\{\delta^\star_w\} \times \{\eta^\star_{xw}\}$ are just the combination of the extreme points of each space. Some combinations $\delta^\star_x \times \eta^\star_{xw}$ map to the same $\zeta^\star_{yx.w}$, while the mapping from a given $\delta^\star_x \times \eta^\star_{xw}$ to $\eta^\star_{xw}$ is just the trivial projection. At this stage, we obtain all the extreme points of the polytope $\{\zeta^\star_{yx.w}\} \times \{\eta^\star_{xw}\}$ that are entailed by the factorization of $P(W, X, Y, U)$ and our constraints.

**Step 3** *Using the extreme points of the joint space $\{\zeta^\star_{yx.w}\} \times \{\eta^\star_{xw}\}$, find the dual polytope of this space in terms of linear inequalities. Points in this polytope are convex combinations of $\{\zeta^\star_{yx.w}\} \times \{\eta^\star_{xw}\}$, shown by Dawid (2003) to correspond to the marginalizations over arbitrary $P(U)$. This results in constraints over $\{\zeta_{yx.w}\} \times \{\eta_{xw}\}$.*

This is the core step in Dawid (2003): points in the polytope $\{\zeta^\star_{yx.w}\} \times \{\eta^\star_{xw}\}$ correspond to different marginalizations of $U$ according to different $P(U)$. Describing the polytope in terms of inequalities provides all feasible distributions that result from marginalizing $U$ according to some $P(U)$. Because we included both $\zeta^\star_{yx.w}$ and $\eta^\star_{xw}$ in the same space, this will tie together $P(Y, X \,|\, W)$ and $P(Y \,|\, do(X), W)$.

**Step 4** *Finally, maximize/minimize (4) with respect to $\{\eta_{xw}\}$ subject to the constraints found in Step 3 to obtain upper/lower bounds on the ACE.*

Allowing for the case where $\epsilon_x < 1$ or $\epsilon_y < 1$ is just a matter of changing the first step, where box constraints are set on each individual parameter as a function of the known $P(Y = y, X = x \,|\, W = w)$, prior to the mapping in Step 2. The resulting constraints are now implicitly non-linear in $P(Y = y, X = x \,|\, W = w)$, but at this stage this does not matter as the distribution of the observables is treated as a constant. That is, each resulting constraint in Step 3 is a linear function of $\{\eta_{xw}\}$ and a multilinear function on $\{\{\zeta_{yx.w}\}, \epsilon_x, \epsilon_y, \epsilon_w, \bar{\beta}, \underline{\beta}, P(W)\}$, as discussed in Section 4. Within the objective function (4), the only decision variables are $\{\eta_{xw}\}$, and hence Step 4 still sets up a linear programming problem even if there are multiplicative interactions between $\{\zeta_{yx.w}\}$ and parameters of constraints.

To allow for the case $\underline{\beta} < 1 < \bar{\beta}$, we substitute every occurrence of $\zeta_{yx.w}$ in the constraints by $\kappa_{yx.w} \equiv \sum_U \zeta^\star_{yx.w}P(U)$; notice the difference between $\kappa_{yx.w}$ and $\zeta_{yx.w}$. Likewise, we substitute every occurrence of $\eta_{xw}$ in the constraints by $\omega_{xw} \equiv \sum_U \eta^\star_{xw}P(U)$. Instead of plugging in constants for the values of $\kappa_{yx.w}$ and turning the crank of a linear programming solver, we treat $\{\kappa_{yx.w}\}$ (and $\{\omega_{xw}\}$) as unknowns, linking them to observables and $\eta_{xw}$ by the constraints

$$\begin{matrix} \kappa_{yx.w} \leq \zeta_{yx.w}/\underline{\beta} & \kappa_{yx.w} \geq \zeta_{yx.w}/\bar{\beta} \\ \omega_{xw} \leq \eta_{xw}/\underline{\beta} & \omega_{xw} \geq \eta_{xw}/\bar{\beta} \end{matrix} \tag{9}$$

**input** : Binary data matrix $\mathcal{D}$; set of relaxation parameters $\aleph$; covariate index set $\mathcal{W}$; cause-effect indices $X$ and $Y$

**output**: A set of triplets $(W, \mathbf{Z}, \mathcal{B})$, where $(W, \mathbf{Z})$ is a witness-admissible set pair contained in $\mathcal{W}$ and $\mathcal{B}$ is a distribution over lower/upper bounds on the ACE implied by the pair

**1** $\mathcal{R} \leftarrow \emptyset$;
**2** **for** *each $W \in \mathcal{W}$* **do**
**3**      **for** *every admissible set $\mathbf{Z} \subseteq \mathcal{W}\backslash\{W\}$ identified by $W$ and $\aleph$ given $\mathcal{D}$* **do**
**4**          $\mathcal{B} \leftarrow$ posterior over lower/upper bounds on the ACE as given by $(W, \mathbf{Z}, X, Y, \mathcal{D}, \aleph)$;
**5**          **if** *there is no evidence in $\mathcal{B}$ to falsify the $(W, \mathbf{Z}, \aleph)$ model* **then**
**6**              $\mathcal{R} \leftarrow \mathcal{R} \cup \{(W, \mathbf{Z}, \mathcal{B})\}$;
**7**          **end**
**8**      **end**
**9** **end**
**10** **return** $\mathcal{R}$

**Algorithm 1:** The outline of the Witness Protection Program algorithm.

$$\sum_{yx} \kappa_{yx.w} = 1. \tag{10}$$

Finally, the steps requiring finding extreme points and converting between representations of a polytope can be easily implemented using a package such as Polymake[8] or the scdd package[9] for R. Once bounds are obtained for each particular value of $\mathbf{Z}$, Equation (2) is used to obtain the unconditional bounds assuming $P(\mathbf{Z})$ is known.

In Section 5, we provide some guidance on how to choose the free parameters of the relaxation. However, it is relevant to point out that any choice of $\epsilon_w \geq 0, \epsilon_y \geq 0, \epsilon_x \geq 0, 0 \leq \underline{\beta} \leq 1 \leq \bar{\beta}$ is *guaranteed to provide bounds that are at least as conservative* as the back-door adjusted point estimator of Entner et al. (2013), which is always covered by the bounds. Background knowledge, after a user is suggested a witness and admissible set, can also be used to set relaxation parameters.

So far, the linear programming formulated through Steps 1–4 assumes one has already identified an appropriate witness $W$ and admissible set $\mathbf{Z}$, and that the joint distribution $P(W, X, Y, \mathbf{Z})$ is known. In the next Section, we discuss how this procedure is integrated with statistical inference for $P(W, X, Y, \mathbf{Z})$ and the search procedure of Entner et al. (2013). As the approach provides the witness a degree of protection against faithfulness violations, using a linear program, we call this framework the *Witness Protection Program* (WPP).

### 3.2 Bayesian Learning and Result Summarization

In the previous section, we treated (the conditional) $\zeta_{yx.w}$ and $P(W = w)$ as known. A common practice is to replace them by plug-in estimators (and in the case of a non-empty admissible set $\mathbf{Z}$, an estimate of $P(\mathbf{Z})$ is also necessary). Such models can also be falsified, as

---

8. http://www.poymake.org
9. http://cran.r-project.org/

the constraints generated are typically only supported by a strict subset of the probability simplex. In principle, one could fit parameters without constraints, and test the model by a direct check of satisfiability of the inequalities using the plug-in values. However, this does not take into account the uncertainty in the estimation. For the standard IV model, Ramsahai and Lauritzen (2011) discuss a proper way of testing such models in a frequentist sense.

Our models can be considerably more complicated. Recall that constraints will depend on the extreme points of the $\{\zeta^\star_{yx.w}\}$ parameters. As implied by (6) and (7), extreme points will be functions of $\zeta_{yx.w}$. Writing the constraints fully in terms of the observed distribution will reveal non-linear relationships. We approach the problem in a Bayesian way. We will assume first the dimensionality of $\mathbf{Z}$ is modest (say, 10 or less), as this is the case in most applications of faithfulness to causal discovery. We parameterize $\zeta^{\mathbf{z}}_{yxw} \equiv P(Y = y, X = x, W = w \mid \mathbf{Z} = \mathbf{z})$ as a full $2 \times 2 \times 2$ contingency table[10]. In the context of the linear programming problem of the previous Section, for a given $\mathbf{z}$, we have $\zeta_{yx.w} = \zeta_{yxw}/P(W = w)$, $P(W = w) = \sum_{yx} \zeta_{yxw}$.

Given that the dimensionality of the problem is modest, we assign to each three-variate distribution $P(Y, X, W \mid \mathbf{Z} = \mathbf{z})$ an independent Dirichet prior for every possible assigment of $\mathbf{Z}$, constrained by the inequalities implied by the corresponding polytopes. The posterior is also a 8-dimensional constrained Dirichlet distribution, where we use rejection sampling to obtain a posterior sample by proposing from the unconstrained Dirichlet. A Dirichlet prior is also assigned to $P(\mathbf{Z})$. Using a sample from the posterior of $P(\mathbf{Z})$ and a sample (for each possible value $\mathbf{z}$) from the posterior of $P(Y, X, W \mid \mathbf{Z} = \mathbf{z})$, we obtain a sample upper and lower bound for the ACE by just running the linear program for each sample of $\{\eta^{\mathbf{z}}_{yxw}\}$ and $\{P(\mathbf{Z} = \mathbf{z})\}$.

The full algorithm is shown in Algorithm 1, where $\aleph \equiv \{\epsilon_w, \epsilon_x, \epsilon_y, \underline{\beta}, \bar{\beta}\}$. The search procedure is left unspecified, as different existing approaches can be plugged into this step. See Entner et al. (2013) for a discussion. In Section 6 we deal with small dimensional problems only, using the brute-force approach of performing an exhaustive search for $\mathbf{Z}$. In practice, brute-force can be still valuable by using a method such as discrete PCA (Buntine and Jakulin, 2004) to reduce $\mathcal{W}\backslash\{W\}$ to a small set of binary variables. To decide whether the premises in Rule 1 hold, we merely perform Bayesian model selection with the BDeu score (Buntine, 1991) between the full graph $\{W \to X, W \to Y, X \to Y\}$ (conditional on $\mathbf{Z}$) and the graph with the edge $W \to Y$ removed.

Step 5 in Algorithm 1 is a "falsification test." Since the data might provide a bad fit to the constraints entailed by the model, we opt not to accept every pair $(W, \mathbf{Z})$ that passes Rule 1. One possibility is to calculate the posterior distribution of the model where constraints are enforced, and compare it against the posteriors of the saturated model given by the unconstrained contingency table. This requires another prior over the constraint hypothesis and the calculation of the corresponding marginal likelihoods. As an alternative approach, we adopt the pragmatic rule of thumb suggested by Richardson et al. (2011): sample $M$ samples from the $\{\zeta^{\mathbf{z}}_{yxw}\}$ posterior given the *unconstrained* model, and check the proportion of values that are rejected. If more than 95% of them are rejected, we take this

---

10. That is, we allow for dependence between $W$ and $Y$ given $\{X, \mathbf{Z}\}$, interpreting the decision of independence used in Rule 1 as being only an indicator of approximate independence.

as an indication that the proposed model provides a bad fit and reject the given choice of $(W, \mathbf{Z})$.

The final result provides a set of posterior distributions over bounds, possibly contradictory, which should be summarized as appropriate. One possibility is to check for the union of all intervals or, as a simpler alternative, report the lowest of the lower bound estimates and the highest of the upper bound estimates using a point estimate for each bound[11]:

1. for each $(W, \mathbf{Z})$ in $\mathcal{R}$, calculate the posterior expected value of the lower and upper bounds;

2. report the interval $\mathcal{L} \leq ACE \leq \mathcal{U}$ where $\mathcal{L}$ is the minimum of the lower bounds and $\mathcal{U}$ the maximum of the upper bounds.

Alternatively to using the expected posterior estimator for the lower/upper bounds, one can, for instance, report the 0.025 quantile of the marginal lower bound distribution and the 0.975 quantile of the marginal upper bound distribution. Notice, however, this does not give a 0.95 credible interval over ACE intervals as the lower bound and the upper bound are dependent in the posterior.

In our experiments, we use a different summary. As we calculate the log-marginal posterior $M_1, M_2, M_3, M_4$ for the hypotheses $W \not\perp\!\!\!\perp Y \mid \mathbf{Z}$, $W \perp\!\!\!\perp Y \mid \mathbf{Z}$, $W \perp\!\!\!\perp Y \mid \mathbf{Z} \cup \{X\}$, $W \not\perp\!\!\!\perp Y \mid \mathbf{Z} \cup \{X\}$, respectively, we use the score

$$(M_1 - M_2) + (M_3 - M_4) \tag{11}$$

to assess the quality of the bounds obtained with the corresponding witness-admissible set pair. We then report the corresponding interval and evaluation metric based on this criterion.

### 3.3 A Note on Weak Dependencies

As we briefly mentioned in the previous Section, our parameterization $\{\zeta^{\mathbf{z}}_{yxw}\}$ does not enforce the independence condition $W \perp\!\!\!\perp Y \mid \mathbf{Z} \cup \{X\}$ required by Rule 1. Our general goal is to let WPP accept "near independencies," in which the meaning of the symbol $\perp\!\!\!\perp$ in practice means weak dependence[12]. We do not define what a weak dependence should mean, except for the general guideline that some agreed measure of conditional association should be "small." Our pragmatic view on WPP is that Rule 1, when supported by weak dependencies, should be used as a motivation for the constraints in Section 3.1. That is, the assumption that "weak dependencies are not generated by arbitrary near-path cancellations," reflecting the belief that very weak associations should correspond to weak direct causal effects (and, where this is unacceptable, WPP should either be adapted to exclude

---

11. One should not confuse credible intervals with ACE intervals, as these are two separate concepts: each lower or upper bound is a function of the unknown $P(W, X, Y, \mathbf{Z})$ and needs to be estimated. There is posterior uncertainty over each lower/upper bound as in any problem where a functional of a distribution needs to be estimated. So the posterior distribution and the corresponding *credible intervals* over the *ACE intervals* are perfectly well-defined as in any standard Bayesian inference problem.

12. The procedure that decides conditional independencies in Section 3.2 is a method for testing exact independencies, although the prior on the independence assumption regulates how strong the evidence in the data should be for independence to be accepted.

Figure 3: Posterior over the ACE obtained by three different priors conditioned on a synthetic dataset of size 1,000,000. Posterior computed by running 1,000,000 iterations of Gibbs sampling. The (independent) priors for $\theta^Y_{1.xu}$ and $\theta^X_{x.wu}$ are Beta $(\alpha, \alpha)$, while $\theta^U_u$ is given a Dirichlet $(\alpha, \alpha, \alpha, \alpha)$. We set $\alpha = 0.1, 1, 10$ for the cases shown in (a), (b) and (c), respectively. Vertical red line shows the true ACE, while the population IV bounds are shown with grey lines. As the prior gets less informative (moving from (c) to (a)), the erratic shape of the posterior distribution also shows the effect of bad Gibbs sampling mixing. Even with a very large dataset, the concentration of the posterior is highly dependent on the concentration of the prior.

relevant cases, or not be used). At the same time, users of WPP do not need to accept this view, as the method does not change under the usual interpretation of $\perp\!\!\!\perp$, but computational gains can be obtained by using a parameterization that encodes the independence.

### 3.4 A Note on Unidentifiability

An alternative to bounding the ACE or using back-door adjustments is to put priors directly on the latent variable model for $\{W, X, Y, U\}$. Using the standard IV model as an example, we can define parameters $\theta^Y_{y.xu} \equiv P(Y = y \mid X = x, U = u)$, $\theta^X_{x.wu} \equiv P(X = x \mid W = w, U = u)$ and $\theta^U_u \equiv P(U = u)$, on which priors are imposed[13]. No complicated procedure for generating constraints in the observable marginal is necessary, and the approach provides point estimates of the ACE instead of bounds.

This sounds too good to be true, and indeed it is: results strongly depend on the prior, regardless of sample size. To illustrate this, consider a simulation from a standard IV model (Figure 1(c)), with $\mathbf{Z} = \emptyset$ and $U$ an unobservable discrete variable of 4 levels. We generated a model by setting $P(W = w) = 0.5$ and sampling parameters $\theta^Y_{1.xu}$ and $\theta^X_{1.wu}$ from the uniform $[0, 1]$ distribution, while the 4-dimensional vector $\theta^U_u$ comes from a Dirichlet $(1, 1, 1, 1)$. The resulting model had an ACE of $-0.20$, with a wide IV interval $[-0.50, 0.38]$ as given by the method of Balke and Pearl (1997). Narrower intervals can only

---

13. $P(W = w)$ is not necessary, as the standard IV bounds Balke and Pearl (1997) do not depend on it.

be obtained by making more assumptions: there is no free lunch. However, as in this case where WPP cannot identify any witness, one might put priors on the latent variable model to get a point estimate, such as the posterior expected value of the ACE.

To illustrate the pitfalls of this approach, we perform Bayesian inference by putting priors directly on the CPT parameters of the latent variable model, assuming we know the correct number of levels for $U$. Figure 3 shows some results with a few different choices of priors. The sample size is large enough so that the posterior is essentially entirely within the population bounds and the estimation of $P(W, X, Y, Z)$ is itself nearly exact. The posterior over the ACE covers a much narrower area than the IV interval, but its behaviour is erratic.

This is not to say that informative priors on a latent variable model cannot produce important results. For instance, Steenland and Greenland (2004) discuss how empirical priors on smoking habits among blue-collar workers were used in their epidemiological question: the causal effect of the occupational harzard of silica exposure on lung cancer incidence among industrial sand workers. Smoking is a confounding factor given the evidence that smoking and occupation are associated. The issue was that smoking was unrecorded among the workers, and so priors on the latent variable relationship to the observables were necessary. Notice, however, that this informative prior is essentially a way of performing a back-door adjustment when the adjustment set $\mathbf{Z}$ and treatment-outcome pair $\{X, Y\}$ are not simultaneously measured within the same subjects. When latent variables are "unknown unknowns," a prior on $P(Y \mid X, U)$ may be hard to justify. Richardson et al. (2011) discuss more issues on priors over latent variable models as a way of obtaining ACE point estimates, one alternative being the separation of identifiable and unindentifiable parameters to make transparent the effect of prior (mis)specification.

## 4. Algebraic Bounds and the Back-substitution Algorithm

Posterior sampling is expensive within the context of Bayesian WPP: constructing the dual polytope for possibly millions of instantiations of the problem is time consuming, even if each problem is small. Moreover, the numerical procedure described in Section 3 does not provide any insight on how the different free parameters $\{\epsilon_w, \epsilon_x, \epsilon_y, \underline{\beta}, \bar{\beta}\}$ interact to produce bounds, unlike the analytical bounds available in the standard IV case. Ramsahai (2012) derives analytical bounds under (5) given a *fixed, numerical* value of $\epsilon_w$. We know of no previous analytical bounds as an algebraic function of $\epsilon_w$.

### 4.1 Algebraic Bounds

We derive a set of bounds, whose validity are proved by three theorems. The first theorem derives separate upper and lower bounds on $\omega_{xw}$ using all the assumptions except Equation (5); this means constraints which do not link distributions under different values of $W = w$. The second theorem derives linear constraints on $\{\omega_{xw}\}$ using (5) and more elementary constraints. Our final result will construct less straightforward bounds, again using Equation (5) as the main assumption. As before, assume we are implicitly conditioning on some $\mathbf{Z} = \mathbf{z}$ everywhere.

We introduce the notation

$$
\begin{aligned}
L_{xw}^{YU} &\equiv \max(P(Y=1|X=x,W=w)-\epsilon_y,0)\\
U_{xw}^{YU} &\equiv \min(P(Y=1|X=x,W=w)+\epsilon_y,1)\\
L_{w}^{XU} &\equiv \max(P(X=1|W=w)-\epsilon_x,0)\\
U_{w}^{XU} &\equiv \min(P(X=1|W=w)+\epsilon_x,1)
\end{aligned}
$$

and define $\underline{L}\equiv\min\{L_{xw}^{YU}\},\bar{U}\equiv\max\{U_{xw}^{YU}\}$. Morever, some further redundant notation is used to simplify the description of the constraints:

$$
\begin{aligned}
\delta_{1.w}^{\star} &\equiv \delta_{w}^{\star}\\
\delta_{0.w}^{\star} &\equiv 1-\delta_{w}^{\star}\\
L_{11}^{XU} &\equiv L_{1}^{XU}\\
L_{01}^{XU} &\equiv 1-U_{1}^{XU}\\
U_{11}^{XU} &\equiv U_{1}^{XU}\\
U_{01}^{XU} &\equiv 1-L_{1}^{XU}
\end{aligned}
$$

and, following Ramsahai (2012), for any $x\in\{0,1\}$, we define $x'$ as the complementary binary value (i.e. $x'=1-x$). The same convention applies to pairs $\{w,w'\}$. Finally, define $\chi_{x.w}\equiv\sum_U P(X=x\mid W=w,U)P(U)=\kappa_{1x.w}+\kappa_{0x.w}$.

**Theorem 2** *The following constraints are entailed by the assumptions expressed in Equations (6), (7) and (8):*

$$
\omega_{xw}\leq\min\begin{cases}\kappa_{1x.w}+U_{xw}^{YU}(\kappa_{0x'.w}+\kappa_{1x'.w})\\ \kappa_{1x.w}/L_{xw}^{XU}\\ 1-\kappa_{0x.w}/U_{xw}^{XU}\end{cases} \tag{12}
$$

$$
\omega_{xw}\geq\max\begin{cases}\kappa_{1x.w}+L_{xw}^{YU}(\kappa_{0x'.w}+\kappa_{1x'.w})\\ \kappa_{1x.w}/U_{xw}^{XU}\\ 1-\kappa_{0x.w}/L_{xw}^{XU}\end{cases} \tag{13}
$$

**Theorem 3** *The following constraints are entailed by the assumptions expressed in Equations (5), (6), (7) and (8):*

$$
\omega_{xw}\leq\min\begin{cases}(\kappa_{1x.w'}+\epsilon_w(\kappa_{0x.w'}+\kappa_{1x.w'}))/L_{xw'}^{XU}\\ 1-(\kappa_{0x.w'}-\epsilon_w(\kappa_{0x.w'}+\kappa_{1x.w'}))/U_{xw'}^{XU}\end{cases} \tag{14}
$$

$$
\omega_{xw}\geq\max\begin{cases}(\kappa_{1x.w'}-\epsilon_w(\kappa_{0x.w'}+\kappa_{1x.w'}))/U_{xw'}^{XU}\\ 1-(\kappa_{0x.w'}+\epsilon_w(\kappa_{0x.w'}+\kappa_{1x.w'}))/L_{xw'}^{XU}\end{cases} \tag{15}
$$

$$
\begin{aligned}
\omega_{xw}-\omega_{xw'}U_{x'w}^{XU} &\leq \kappa_{1x.w}+\epsilon_w(\kappa_{0x'.w}+\kappa_{1x'.w})\\
\omega_{xw}-\omega_{xw'}L_{x'w}^{XU} &\geq \kappa_{1x.w}-\epsilon_w(\kappa_{0x'.w}+\kappa_{1x'.w})\\
\omega_{xw}-\omega_{xw'}U_{x'w}^{XU} &\geq 1-\kappa_{0x.w}-U_{x'w}^{XU}-\epsilon_w(\kappa_{0x'.w}+\kappa_{1x'.w})\\
\omega_{xw}-\omega_{xw'}L_{x'w}^{XU} &\leq 1-\kappa_{0x.w}-L_{x'w}^{XU}+\epsilon_w(\kappa_{0x'.w}+\kappa_{1x'.w})\\
\omega_{xw}-\omega_{xw'} &\leq \epsilon_w\\
\omega_{xw}-\omega_{xw'} &\geq -\epsilon_w
\end{aligned} \tag{16}
$$

**Theorem 4** *The following constraints are entailed by the assumptions expressed in Equations (5), (6), (7) and (8):*

$$\omega_{xw} \leq \min \begin{cases} \kappa_{1x'.w'} + \kappa_{1x.w'} + \kappa_{1x.w} - \kappa_{1x'.w} + \chi_{x'w}(\bar{U} + \underline{L} + 2\epsilon_w) - \underline{L} \\ \kappa_{1x'.w} + \kappa_{1x.w} + \kappa_{1x.w'} - \kappa_{1x'.w'} + 2\chi_{x'w}\epsilon_w + \chi_{x'w'}(\bar{U} + \underline{L}) - \underline{L} \end{cases} \tag{17}$$

$$\omega_{xw} \geq \max \begin{cases} -\kappa_{1x'.w'} + \kappa_{1x.w'} + \kappa_{1x'.w} + \kappa_{1x.w} + \chi_{x'w'}(\bar{U} + \underline{L}) - 2\epsilon_w\chi_{x'w} - \bar{U} \\ -\kappa_{1x'.w} + \kappa_{1x.w} + \kappa_{1x'.w'} + \kappa_{1x.w'} - \chi_{x'w}(2\epsilon_w - \bar{U} - \underline{L}) - \bar{U} \end{cases} \tag{18}$$

$$\begin{aligned}
\omega_{xw} + \omega_{x'w} - \omega_{x'w'} &\geq \kappa_{1x'.w} + \kappa_{1x.w} - \kappa_{1x'.w'} + \kappa_{1x.w'} - \chi_{xw'}(\bar{U} + \underline{L} + 2\epsilon_w) + \underline{L} \\
\omega_{xw} + \omega_{x'w'} - \omega_{x'w} &\geq \kappa_{1x'.w'} + \kappa_{1x.w'} - \kappa_{1x'.w} + \kappa_{1x.w} - 2\chi_{xw}\epsilon_w - \chi_{xw}(\bar{U} + \underline{L}) + \underline{L} \\
\omega_{xw} + \omega_{x'w'} - \omega_{x'w} &\leq -\kappa_{1x'.w} + \kappa_{1x.w} + \kappa_{1x'.w'} + \kappa_{1x.w'} - \chi_{xw}(\bar{U} + \underline{L}) + 2\epsilon_w\chi_{xw} + \bar{U} \\
\omega_{xw} + \omega_{x'w} - \omega_{x'w'} &\leq -\kappa_{1x'.w'} + \kappa_{1x.w'} + \kappa_{1x'.w} + \kappa_{1x.w} + \chi_{xw'}(2\epsilon_w - \bar{U} - \underline{L}) + \bar{U}
\end{aligned} \tag{19}$$

Although at first sight such relations seem considerably more complex than those given by Ramsahai (2012), on closer inspection they illustrate qualitative aspects of our free parameters. For instance, consider

$$\omega_{xw} \geq \kappa_{1x.w} + L_{xw}^{YU}(\kappa_{0x'.w} + \kappa_{1x'.w}),$$

one of the instances of (13). If $\epsilon_y = 1$ and $\underline{\beta} = \bar{\beta} = 1$, then $L_{xw}^{YU} = 0$ and this relation collapses to $\eta_{xw} \geq \zeta_{1x.w}$, one of the original relations found by Balke and Pearl (1997) for the standard IV model. Decreasing $\epsilon_y$ will linearly increase $L_{xw}^{YU}$ only after $\epsilon_y \leq P(Y = 1 \mid X = x, W = w)$, tightening the corresponding lower bound given by this equation.

Consider now

$$\omega_{xw} \leq 1 - (\kappa_{0x.w'} - \epsilon_w(\kappa_{0x.w'} + \kappa_{1x.w'}))/U_{xw'}^{XU}.$$

If also $\epsilon_w = 0$ and $\epsilon_x = 1$, from this inequality it follows that $\eta_{xw} \leq 1 - \zeta_{0x.w'}$. This is another of the standard IV inequalities (Balke and Pearl, 1997).

Equation (5) implies $|\omega_{x'w} - \omega_{x'w'}| \leq \epsilon_w$, and as such by setting $\epsilon_w = 0$ we have that

$$\omega_{xw} + \omega_{x'w} - \omega_{x'w'} \geq \kappa_{1x'.w} + \kappa_{1x.w} - \kappa_{1x'.w'} + \kappa_{1x.w'} - \chi_{xw'}(\bar{U} + \underline{L} + 2\epsilon_w) + \underline{L} \tag{20}$$

implies $\eta_{xw} \geq \eta_{1x.w} + \eta_{1x.w'} - \eta_{1x'.w'} - \eta_{0x.w'}$, one of the most complex relationships in (Balke and Pearl, 1997). Further geometric intuition about the structure of the binary standard IV model is given by Richardson and Robins (2010).

These bounds are not tight, in the sense that we opt not to fully exploit all possible algebraic combinations for some results, such as (20): there we use $\underline{L} \leq \eta_{xw}^\star \leq \bar{U}$ and $0 \leq \delta_w^\star \leq 1$ instead of all possible combinations resulting from (6) and (7). The proof idea in Appendix A can be further refined, at the expense of clarity. Because our derivation is a further relaxation, our final bounds are more conservative (i.e., looser).

## 4.2 Efficient Optimization and Falsification Tests

Besides providing insight into the structure of the problem, the algebraic bounds give an efficient way of checking whether a proposed parameter vector $\{\zeta_{yxw}\}$ is valid in Step 5 of Algorithm 1, as well as finding the ACE bounds: we can now use back-substitution on the symbolic set of constraints to find box constraints $\mathcal{L}_{xw} \leq \omega_{xw} \leq \mathcal{U}_{xw}$. The proposed parameter will be rejected whenever an upper bound is smaller than a lower bound, and (4) can be trivially optimized conditioning only on the box constraints—this is yet another relaxation, added on top of the ones used to generate the algebraic inequalities. We initialize by intersecting all algebraic box constraints (of which (12) and (14) are examples); next we refine these by scanning relations $\pm\omega_{xw} - a\omega_{xw'} \leq c$ (the family given by (16)) in lexicographical order, and tightening the bounds of $\omega_{xw}$ using the current upper and lower bounds on $\omega_{xw'}$ where possible. We then identify constraints $\mathcal{L}_{xww'} \leq \omega_{xw} - \omega_{xw'} \leq \mathcal{U}_{xww'}$ starting from $-\epsilon_w \leq \omega_{xw} - \omega_{xw'} \leq \epsilon_w$ and the existing bounds, and plug them into relations $\pm\omega_{xw} + \omega_{x'w} - \omega_{x'w'} \leq c$ (as exemplified by (20)) to get refined bounds on $\omega_{xw}$ as functions of $(\mathcal{L}_{x'ww'}, \mathcal{U}_{x'ww'})$. We iterate this until convergence, which is guaranteed since lower/upper bounds never decrease/increase at any iteration. This back-substitution of inequalities follows the spirit of message-passing and it can be orders of magnitude more efficient than the fully numerical solution, while not increasing the width of the intervals by too much. In Section 6, we provide evidence for this claim. The back-substitution method is t used in our experiments, combined with the fully numerical linear programming approach as explained in Section 6. The full algorithm is given in Algorithm 2.

## 5. Choosing Relaxation Parameters

The free parameters $\aleph \equiv \{\epsilon_w, \epsilon_x, \epsilon_y, \underline{\beta}, \bar{\beta}\}$ do not have an unique, clear-cut, domain-free procedure by which they can be calibrated. However, as we briefly discussed in Section 3, it is useful to state explicitly the following worst-case scenario guarantee of WPP:

**Corollary 5** *Given $W \not\perp\!\!\!\perp Y \mid \mathbf{Z}$ and $W \perp\!\!\!\perp Y \mid \{X, \mathbf{Z}\}$, the* WPP *population bounds on the ACE will always include the back-door adjusted population ACE based on* $\mathbf{Z}$.

**Proof** The proof follows directly by plugging in the quantities $\epsilon_w = \epsilon_y = \epsilon_x = 0$, $\underline{\beta} = \bar{\beta} = 1$, into the analytical bounds of Section 4.1, which will give the tightest bounds on the ACE (generalized to accommodate a background set $\mathbf{Z}$): a single point, which also happens to be the functional obtained by the back-door adjustment. ∎

The implication is that, regardless of the choice of free parameters, the result is guaranteed to be more conservative than the one obtained using the faithfulness assumption. In any case, this does not mean that a judicious choice of relaxation parameters is of secondary importance.

The setting of relaxation parameters can be interpreted in two ways:

- $\aleph$ is set prior to calculating the ACE; this uses expert knowledge concerning the remaining amount of unmeasured confounding, decided with respect to the provided admissible set and witness, or by a default rule concerning beliefs on faithfulness violations;

**input** : Distributions $\{\zeta_{yx.w}\}$ and $\{P(W = w)\}$;
**output**: Lower and upper bounds $(\mathcal{L}_{xw}, \mathcal{U}_{xw})$ for every $\omega_{xw}$

**1** Find tightest lower and upper bounds $(\mathcal{L}_{xw}, \mathcal{U}_{xw})$ for each $\omega_{xw}$ using inequalities (12), (13) (14), (15), (17) and (18);

**2** Let $\mathcal{L}_{xw}^{\epsilon_w}$ and $\mathcal{U}_{xw}^{\epsilon_w}$ be lower/upper bounds of $\omega_{xw} - \omega_{xw'}$;

**3 for** *each pair $(x, w) \in \{0,1\}^2$* **do**

**4** $\quad$ $\mathcal{L}_{xw}^{\epsilon_w} \leftarrow -\epsilon_w$;

**5** $\quad$ $\mathcal{U}_{xw}^{\epsilon_w} \leftarrow \epsilon_w$;

**6 end**

**7 while** *TRUE* **do**

**8** $\quad$ **for** *each relation $\omega_{xw} - b \times \omega_{xw'} \le c$ in (16)* **do**

**9** $\quad\quad$ $\mathcal{U}_{xw}^{\epsilon_w} \leftarrow \min\{\mathcal{U}_{xw}^{\epsilon_w}, (b-1)\mathcal{L}_{xw} + c\}$

**10** $\quad$ **end**

**11** $\quad$ **for** *each relation $\omega_{xw} - b \times \omega_{xw'} \ge c$ in (16)* **do**

**12** $\quad\quad$ $\mathcal{L}_{xw}^{\epsilon_w} \leftarrow \max\{\mathcal{L}_{xw}^{\epsilon_w}, (b-1)\mathcal{U}_{xw} + c\}$

**13** $\quad$ **end**

**14** $\quad$ **for** *each relation $\omega_{xw} + \omega_{x'w} - \omega_{x'w'} \le c$ in (19)* **do**

**15** $\quad\quad$ $\mathcal{U}_{xw} \leftarrow \min\{\mathcal{U}_{xw}, c - \mathcal{L}_{xw'}^{\epsilon_w}\}$

**16** $\quad$ **end**

**17** $\quad$ **for** *each relation $\omega_{xw} - (\omega_{x'w} - \omega_{x'w'}) \le c$ in (19)* **do**

**18** $\quad\quad$ $\mathcal{U}_{xw} \leftarrow \min\{\mathcal{U}_{xw}, c + \mathcal{U}_{xw'}^{\epsilon_w}\}$

**19** $\quad$ **end**

**20** $\quad$ **for** *each relation $\omega_{xw} + \omega_{x'w} - \omega_{x'w'} \ge c$ in (19)* **do**

**21** $\quad\quad$ $\mathcal{U}_{xw} \leftarrow \max\{\mathcal{U}_{xw}, c - \mathcal{U}_{xw'}^{\epsilon_w}\}$

**22** $\quad$ **end**

**23** $\quad$ **for** *each relation $\omega_{xw} - (\omega_{x'w} - \omega_{x'w'}) \ge c$ in (19)* **do**

**24** $\quad\quad$ $\mathcal{U}_{xw} \leftarrow \max\{\mathcal{U}_{xw}, c + \mathcal{L}_{xw'}^{\epsilon_w}\}$

**25** $\quad$ **end**

**26** $\quad$ **if** *no changes in $\{(\mathcal{L}_{xw}, \mathcal{U}_{xw})\}$* **then**

**27** $\quad\quad$ break

**28** $\quad$ **end**

**29 end**

**30 return** $(\mathcal{L}_{xw}, \mathcal{U}_{xw})$ *for each $(x, w) \in \{0,1\}^2$*

**Algorithm 2:** The iterative back-substitution procedure for bounding $\mathcal{L}_{xw} \le \omega_{xw} \le \mathcal{U}_{xw}$ for all combinations of $x$ and $w$ in $\{0,1\}^2$.

- ℵ is deduced by the outcome of a sensivity analysis procedure; given a particular interval length $L$, we derive a quantification of faithfulness violations (represented by ℵ) required to generate causal models compatible with the observational data and an interval of length $L$ containing the ACE;

That is, in the first scenario the input is ℵ, the output are bounds on the ACE. In the second scenario, the input is the acceptable width of an interval containing the ACE, the output are the bounds on the ACE *and* a choice of ℵ. In his rejoinder to the discussion

of (Rosenbaum, 2002b), Rosenbaum points out that the sensitivity analysis procedure just states the logical outcome of the structural assumptions: the resulting deviation of, say, $P(Y = 1 \mid X = x, W = w)$ from $P(Y = 1 \mid X = x, W = w, U = u)$ required to explain the given length of variation on the ACE is not directly imposed by expert knowledge concerning confounding effects. Expert knowledge is of course still necessary to decide whether the resulting deviation is unlikely or not (and hence, whether the resulting interval is believable), although communication by sensitivity analysis might facilitate discussion and criticism of the study.

Motivated by the idea of starting from a pre-specified length $L$ for the resulting interval around the ACE, in what follows we describe two possible ways of setting relaxation parameters. We contrast the methods against the idea of putting priors on latent variable models, as discussed in Section 3.4.

## 5.1 Choice by Grid Search Conditioned on Acceptable Information Loss

One pragmatic default rule is to first ask how wide an ACE interval can be so that the result is still useful for the goals of the analysis (e.g., sorting possible controls $X$ as candidates for a lab experiment based on lower bounds on the ACE). Let $L$ be the interval width the analyst is willing to pay for. Set $\epsilon_w = \epsilon_x = \epsilon_y = k_\epsilon$ and $\underline{\beta} = c$, $\bar{\beta} = 1/c$, for some pair $(k_\epsilon, c)$ such that $0 \leq k < 1$, $0 < c \leq 1$, and let $(k, c)$ range over a grid of values. For each witness/admissible set candidate pair, pick the $(k, c)$ choice(s) entailing interval(s) of length closest to $L$. In case of more than one solution, summarize them by a criterion such the union of the intervals.

This methodology provides an explicit trade-off between length of the interval and tightness of assumptions. Notice that, starting from the backdoor-adjusted point estimator of Entner et al. (2013), it is not clear how one would build a procedure to provide such a trade-off: that is, a procedure by which one could build an interval around the point estimate within a given acceptable amount of information loss. WPP provides a principled way of building such an interval, with the resulting assumptions on $\aleph$ being explicitly revealed as a by-product. If the analyst believes that the resulting values of $\aleph$ are not strict enough, and no substantive knowledge exists that allows particular parameters to be tightened up, then one either has to concede that wider intervals are necessary or to find other means of identifying the ACE unrelated to the faithfulness assumption.

In the experiments in Section 6.2, we define a parameter space of $k_\epsilon \in \{0.05, 0.10, \ldots, 0.30\}$ and $c \in \{0.9, 1\}$. More than one interval of approximately the same width are identified. For instance, the configurations $(k_\epsilon = 0.25, c = 1)$ and $(k_\epsilon = 0.05, c = 0.9)$ both produce intervals of approximately length 0.30.

## 5.2 Linking Selection on the Observables to Selection on the Unobservables

The trade-off framework assumes the analyst has a known tolerance level for information loss (that is, the length of the interval around the back-door adjusted estimator), around which an automated procedure for choosing $\aleph$ can be constructed. Alternatively, one might choose a value of $\aleph$ *a priori* using information from the problem at hand, and accept the information loss that it entails. This still requires a way of connecting prior assumptions to data.

Observational studies cannot be carried out without making assumptions that are untestable given the data at hand. There will always be degrees of freedom that must be chosen, even if such choices are open to criticism. The game is to provide a language to express assumptions in as transparent a manner as possible. Our view on priors for the latent variable model (Section 3.4) is that such prior knowledge is far too difficult to justify when the interpretation of $U$ is unclear. Moreover, putting a prior on a parameter such as $P(Y = 1 \mid X = x, W = w, U = u)$ so that this prior is bounded by the constraint $|P(Y = 1 \mid X = x, W = w, U = u) - P(Y = 1 \mid X = w, W = w)| \le \epsilon_w$ has no clear advantage over the WPP: a specification of the shape of this prior is still necessary and may have undesirable side effects; it has no computational advantages over the WPP, as constraints will have to be dealt with now within a Markov chain Monte Carlo procedure; it provides no insight on how constraints are related to one another (Section 4); it still suggests a point estimate that should not be trusted lightly, and posterior bounds which cannot be interpreted as worst-case bounds; and it still requires a choice of $\epsilon_w$.

That is not to say that subjective priors on the relationship between $U$ and the observables cannot be exploited, but the level of abstraction at which they need to be specified should have advantages when compared to the latent variable model approach. For instance, Altonji et al. (2005) introduced a framework to deal with violations of the IV assumptions (in the context of linear models). Their main idea is to linearly decompose the (observational) dependence of $W$ and $\mathbf{Z}$, and the (causal) dependence of $Y$ and $\mathbf{Z}$, as two signal-plus-noise decompositions, and assume that dependence among the signals allows one to infer the dependence among the noise terms. In this linear case, the dependence among noise terms gives the association between $W$ and $Y$ through unmeasured confounders. The constraint given by the assumption can then be used to infer bounds on the (differential) ACE. The details are not straightforward, but the justification for the assumption is indirectly derived by assuming $\mathbf{Z}$ is chosen by a sampling mechanism that picks covariates from the space of confounders $U$, so that $|\mathbf{Z}|$ and $|U|$ are large. The principal idea is that the dependence between the covariates which are observed (i.e. $\mathbf{Z}$) and the other variables $(W, X, Y)$ should tell us something about the impact of the unmeasured confounders. Their method is presented for linear models only, and the justification requires a very large $|\mathbf{Z}|$.

We introduce a very different method inspired by the same general principle, but exploiting the special structure of our procedure. Instead of relying on linearity and a fixed set of covariates, consider the following postulate: the variability of back-door adjusted ACE estimators based on different admissible sets, as implied by Rule 1, should provide some information about the extent of the violations of faithfulness in the given domain.

For simplicity of exposition, we adopt the parameterization of $\aleph$ as given by the three parameters $(\epsilon_w, \epsilon_{xy} = \epsilon_x = \epsilon_y, \beta = \underline{\beta} = 1/\bar{\beta})$. Given a prior $\pi(\epsilon_w, \epsilon_{xy}, \beta)$ over the three-dimensional unit cube $[0, 1]^3$, we want to assess probable values of such parameters using a "likelihood" function that explains the variability of the ACEs provided by Entner et al.'s rule. We want the posterior to converge to the single values $\epsilon_w = 0, \epsilon_{xy} = 0$ and $\beta = 1$ as the number of witness/admissible set pairs increase and under the condition that they agree on the same value.

For that, we will consider a *target* witness/admissible set pair $(W^\star, \mathbf{Z}^\star)$, and a *reference set* $\mathcal{R}$ of other admissible sets. Given $\aleph$, $W^\star$, $\mathbf{Z}^\star$ and the joint distribution over observables $P(\mathbf{V})$, a lower bound $LB^\star$ and an upper bound $UB^\star$ on the ACE are determined. Given

the bounds, we define a likelihood function

$$\mathcal{L}(\aleph; P(\mathbf{V}), W^\star, \mathbf{Z}^\star, \mathcal{R}) \equiv \prod_{i=1}^{n} p_{N[-1,1]}(ACE_i; m(LB^\star, UB^\star), v(LB^\star, UB^\star)) \qquad (21)$$

where $p_{N[-1,1]}(\cdot; m, v)$ is a truncated Gaussian density on $[-1, 1]$ proportional to a Gaussian with mean $m$ and variance $v$; $m(LB^\star, UB^\star)$ and $v(LB^\star, UB^\star)$ are functions of the bounds. Along with the prior, this defines a posterior over $\aleph$; $ACE_i$ is the back-door adjusted ACE obtained with the $i$th entry of $\mathcal{R}$, conditioned on $P(\mathbf{V})$.

There are many degrees of freedom in this formulation, and we do not claim it represents anything other than subjective knowledge: an approximation to the idea that large/small variability of the ACEs should indicate large/small violations of faithfulness, and that we should get more confident about the magnitude of the violations as more ACEs are reported by Entner et al.'s back-door estimator. In our implementation we treat $m$ as a free parameter, with a uniform prior in $[LB^\star, UB^\star]$. We treat $v$ as a deterministic function of the bounds,

$$v(LB^\star, UB^\star) \equiv ((UB^\star - LB^\star)/6)^2 \qquad (22)$$

to reflect the assumption that the interval $[LB^\star, UB^\star]$ should cover a large amount of mass of the model—in this case, $UB^\star - LB^\star$ is approximately 6 times the standard deviation of the likelihood model.

Finally, our problem has one last degree of freedom: (21) treats the ACEs implied by $\mathcal{R}$ as conditionally independent. Since many admissible sets overlap, this can result in over-confident posteriors, in the sense that they do not reflect our belief that similar admissible sets do not provide independent pieces of evidence concerning violations of faithfulness. Our pragmatic correction to that is to discard from $\mathcal{R}$ any admissible set which is a strict superset of some other element of $\mathcal{R} \cup \{\mathbf{Z}^\star\}$. Notice that in some situations, $\mathcal{R}$ might contain the empty set as a possible admissible set, implying that the resulting $\mathcal{R}$ will contain at most one element (the empty set itself). Optionally, one might forbid *a priori* the empty set ever entering $\mathcal{R}$.

The criterion above can be refined in many ways: among other issues, one does not want to inflate the confidence on $\aleph$ by measuring many highly correlated (sets of) covariates that will end up being added independently to $\mathcal{R}$. One idea is to modify the likelihood function to allow for dependencies among different ACE "data points." We leave this as future work.

Besides the priors over $\aleph$ and $m(LB^\star, UB^\star)$, we can also in principle define a prior for $P(\mathbf{V})$. In the following illustration, and in the application in Section 6.3, we simplify the analysis by treating $P(\mathbf{V})$ as known, using the posterior expected value of $P(\mathbf{V})$ given an BDeu prior with effective sample size of 10. The full algorithm is shown in Algorithm 3.

It should be stressed out that the posterior over $\aleph$ will in general be unidentifiable, since the sufficient statistics for the likelihood are the upper and lower bounds and different values of $\aleph$ can yield the same bounds. Our implementation of Algorithm 3 consists of using a simple Metropolis-Hastings scheme to sample each of the three components $\epsilon_w, \epsilon_{xw}, \beta$ one at a time, and mixing will be a practical issue. Priors will matter. In particular, a situation with a very small $\mathcal{R}$ and an uniform prior $\pi(\aleph)$ might require many MCMC iterations.

Consider Figure 4. Here, we have a synthetic problem where we know no admissible set exists. Due to sampling variability and near-faithfulness violations, WPP identifies

**input** : Data set $\mathcal{D}$ over observed variables $\mathbf{V}$; hyperparameter $\alpha$ for the BDeu prior; prior $\pi(\aleph)$; a flag `allow_empty` indicating whether empty sets are allowed

**output**: A posterior distribution over $\aleph$

**1** Find all witness/admissible set pairs $\mathcal{P}$ according to Rule 1, data $\mathcal{D}$ and BDeu hyperparameter $\alpha$

**2** Let $(W^\star, \mathbf{Z}^\star)$ be the highest scoring pair according to the WPP scoring rule (11)

**3** Let $\mathcal{R}$ be the set of all admissible sets in $\mathcal{P}$

**4** Remove the empty set from $\mathcal{R}$ if `allow_empty` is false

**5** Remove from $\mathcal{R}$ any set that strictly contains some other set in $\mathcal{R}$

**6** Remove $\mathbf{Z}^\star$ from $\mathcal{R}$

**7** Let $P(\mathbf{V})$ be the posterior expected value of the distribution of $\mathcal{V}$ as given by $\mathcal{D}$ and $\alpha$

**8** Return the posterior distribution implied by $\pi(\aleph)$, $\mathcal{R}$, $(W^\star, \mathbf{Z}^\star)$ and $P(\mathbf{V})$

**Algorithm 3:** Finding a posterior distribution over relaxation parameters $\aleph$ using candidate solutions generated by Entner et al.'s Rule 1.

three such sets. This is one of the hardest positions for the $\aleph$ learning procedure, since the posterior will also be very broad. The true ACE is $-0.16$, while the estimated ACEs given by $\mathcal{R}$ are $\{-0.44, -0.34\}$. With only two (reasonably spread out) data points and an uniform prior for $\aleph$, we obtain the posterior distribution for the entries of $\aleph$ as shown in Figure 4(a). This reflects uncertainty and convergence difficulties of the MCMC procedure. More informative priors make a difference, as shown in Figure 4(b). In Section 6.3, a simple empirical study with far more concentrated ACEs provides a far more tightly concentrated set of marginal posteriors.

## 6. Experiments

In this Section, we start with a comparison of the back-substitution algorithm of Section 4.2 against the fully numerical procedure, which generates constraints using standard algorithms for changing between polytope representations. We then perform studies with synthetic data, comparing different back-door estimation algorithms against WPP. Finally, we perform analysis with a real dataset.

### 6.1 Empirical Investigation of the Back-substitution Algorithm

We compare the back-substitution algorithm introduced in Section 4.2 with the fully numerical algorithm. Comparison is done in two ways: (i) computational cost, as measured by the wallclock time taken to generate 100 samples by rejection sampling; (ii) width of the generated intervals. As discussed in Section 4.2, bounds obtained by the back-substitution algorithm are at least as wide as in the numerical algorithm, barring rounding problems[14].

---

14. About 1% of the time we observed numerical problems with the polytope generator, as we were not using rational arithmetic in order to speed it up. Those were excluded from the statistics reported in this Section.

(a)                                                       (b)

Figure 4: In (a), the posterior marginal densities for $\epsilon_w, \epsilon_{xy}$ and $\beta$ with an uniform prior, smoothed kernel density estimates based on $10,000$ Monte Carlo samples. An analogous picture is shown in (b), for the situation where the prior is now a product of three univariate truncated Gaussians in $[0, 1]$, each marginal proportional to an univariate Gaussian with means $(0.2, 0.2, 0.95)$ and variances $(0.1, 0.1, 0.05)$, respectively.

We ran two batches of 1000 trials each, varying the level of the relaxation parameters. In the first batch, we set $\epsilon_x = \epsilon_y = \epsilon_w = 0.2$, and $\underline{\beta} = 0.9$, $\bar{\beta} = 1.1$. In the second batch, we change parameters so that $\underline{\beta} = \bar{\beta} = 1$. Experiments were run on a Intel Xeon E5-1650 at 3.20Ghz. Models were simulated according the the structure $W \rightarrow X \rightarrow Y$, sampling each conditional distribution of a vertex being equal to 1 given its parent from the uniform $(0, 1)$ distribution. The numerical procedure of converting extreme points to linear inequalities was done using the package RCDD, a R wrapper for the *cddlib* by Komei Fukuda. Inference is done by rejection sampling, requiring 100 samples per trial. We fix the number of interations of the back-substitution method to 4, which is more than enough to achieve convergence. All code was written in R.

For the first batch, the average time difference between the fully numerical method and the back-substitution algorithm was 1 second, standard deviation (s.d.) 0.34. The ratio between times had a mean of 203 (s.d. 82). Even with a more specialized implementation of the polytope dualization step[15], two orders of magnitude of difference seem hard to remove by better coding. Concerning interval widths, the mean difference was 0.15 (s.d.

---

15. One advantage of the analytical bounds, as used by the back substitution method, is that it is easy to express them as matrix operations over all Monte Carlo samples, while the polytope construction requires iterations over the samples.

0.06), meaning that the back-substitution on average has intervals where the upper bound minus the lower bound difference is 0.15 units more than the numerical method, under this choice of relaxation parameters and averaged over problems generated according to our simulation scheme. There is a correlation between the width difference and the interval width given by the numerical method the gap, implying that differences tend to be larger when bounds are looser: the gap between methods was as small as 0.04 for a fully numerical interval of width 0.19, and as large as 0.23 for a fully numerical interval of width 0.49. For the case where $\bar{\beta} = \underline{\beta} = 1$, the average time difference was 0.92 (s.d. of 0.24), ratio of 152 (s.d. 54.3), interval width difference of 0.09 (s.d. 0.03); The gap was as small as 0.005 for a fully numerical interval of width 0.09, and as large as 0.17 for a fully numerical interval of with 0.23.

## 6.2 Synthetic Studies

We describe a set of synthetic studies where we assess the trade-off between ACE intervals and error, as wider intervals will be less informative than point estimators such as the back-door adjustment, but by definition have more chances of correctly covering the ACE.

In the synthetic study setup, we compare our method against NE1 and NE2, two naïve point estimators defined by back-door adjustment on the whole of set of available covariates $\mathcal{W}$ and on the empty set, respectively. The former is widely used in practice, even when there is no causal basis for doing so (Pearl, 2009). The point estimator of Entner et al. (2013), based solely on the faithfulness assumption, is also assessed.

We generate problems where conditioning on the whole set $\mathcal{W}$ is guaranteed to give incorrect estimates. In detail: we generate graphs where $\mathcal{W} \equiv \{Z_1, Z_2, \ldots, Z_8\}$. Four independent latent variables $L_1, \ldots, L_4$ are added as parents of each $\{Z_5, \ldots, Z_8\}$; $L_1$ is also a parent of $X$, and $L_2$ a parent of $Y$. $L_3$ and $L_4$ are each randomly assigned to be a parent of either $X$ or $Y$, but not both. $\{Z_5, \ldots, Z_8\}$ have no other parents. The graph over $Z_1, \ldots, Z_4$ is chosen by adding edges uniformly at random according to the lexicographic order. In consequence using the full set $\mathcal{W}$ for back-door adjustment is always incorrect, as at least four paths $X \leftarrow L_1 \rightarrow Z_i \leftarrow L_2 \rightarrow Y$ are active for $i = 5, 6, 7, 8$. The conditional probabilities of a vertex given its parents are generated by a logistic regression model with pairwise interactions, where parameters are sampled according to a zero mean Gaussian with standard deviation 20 / number of parents. Parameter values are also squashed, so that if the generated value if greater than 0.975 or less than 0.025, it is resampled uniformly in $[0.950, 0.975]$ or $[0.025, 0.050]$, respectively.

We analyze two variations: one where it is guaranteed that at least one valid pair witness-admissible set exists; in the other, all latent variables in the graph are set also as common parents also of $X$ and $Y$, so no valid witness exists. We divide each variation into two subcases: in the first, "hard" subcase, parameters are chosen (by rejection sampling, proposing from the model described in the previous paragraph) so that NE1 has a bias of at least 0.1 in the population; in the second, no such a selection exists, and as such our exchangeable parameter sampling scheme makes the problem relatively easy. We summarize each WPP interval by the posterior expected value of the lower and upper bounds. In general WPP returns more than one bound: we select the upper/lower bound corresponding

to the $(W, \mathbf{Z})$ pair which maximizes the score described at the end of Section 3.2. A BDeu prior with an equivalent sample size of 10 was used.

Our main evaluation metric for an estimate is the Euclidean distance (henceforth, "error") between the true ACE and the closed point in the given estimate, whether the estimate is a point or an interval. For methods that provide point estimates (NE1, NE2, and faithfulness), this means just the absolute value of the difference between the true ACE and the estimated ACE. For WPP, the error of the interval $[\mathcal{L}, \mathcal{U}]$ is zero if the true ACE lies in this interval. We report *error average* and *error tail mass at 0.1*, the latter meaning the proportion of cases where the error exceeds 0.1. Moreover, the faithfulness estimator is defined by averaging over all estimated ACEs as given by the accepted admissible sets in each problem.

As discussed in Section 5.1, WPP can be understood as providing a trade-off between information loss and accuracy. For instance, while the trivial interval $[-1, 1]$ will always have zero error, it is not an interesting solution. We assess the trade-off by running simulations at different levels of $k_\epsilon$, where $\epsilon_w = \epsilon_y = \epsilon_x = k_\epsilon$. We also have two configurations for $\{\underline{\beta}, \bar{\beta}\}$: we set them at either $\underline{\beta} = \bar{\beta} = 1$ or $\underline{\beta} = 0.9, \bar{\beta} = 1.1$.

For the cases where no witness exists, Entner's Rule 1 should theoretically report no solution. Entner et al. (2013) used stringent thresholds for deciding when the two conditions of Rule 1 held. Instead we take a more relaxed approach, using a uniform prior on the hypothesis of independence. As such, due to the nature of our parameter randomization, more often than not is will propose at least one witness. That is, for the problems where no exact solution exists, we assess how sensitive the methods are given conclusions taken from "approximate independencies" instead of exact ones.

The analytical bound are combined with the numerical procedure as follows. We use the analytical bounds to test each proposed model using the rejection sampling criterion. Under this scheme, we calculate the posterior expected value of the contingency table and, using this single point, calculate the bounds using the fully numerical method. This is not guaranteed to work: the point estimator using the analytical bounds might lie outside the polytope given by the full set of constraints. If this situation is detected, we revert to calculating the bounds using the analytical method. The gains in interval length reduction using the full numerical method are relatively modest (e.g., at $k_\epsilon = 0.20$, the average interval width reduced from 0.30 to 0.24) but depending on the application they might make a sensible difference.

We simulate 100 datasets for each one of the four cases (hard case/easy case, with theoretical solution/without theoretical solution), 5000 points per dataset, 1000 Monte Carlo samples per decision. Results for the point estimators (NE1, NE2, faithfulness) are obtained using the population contingency tables. Results are summarized in Table 6.2. The first observation is at very low levels of $k_\epsilon$ we increase the ability to reject all witness candidates: this is due mostly not because Rule 1 never fires, but because the falsification rule of WPP (which does not enforce independence constraints) rejects the proposed witnesses found by Rule 1. The trade-off set by WPP is quite stable, where larger intervals are indeed associated with smaller error. The point estimates vary in quality, being particularly bad in the situation where no witness should theoretically exist. The set-up where $\underline{\beta} = 0.9, \bar{\beta} = 1$ is particularly less informative. At $k_\epsilon = 0.2$, we obtain interval widths around 0.50. As

| $k_\epsilon$ | Found | Faith.1 | | WPP1 | | Width1 | WPP2 | | Width2 |
|---|---|---|---|---|---|---|---|---|---|
| **Hard, Solvable:** NE1 $= (0.12, 1.00)$, NE2 $= (0.02, 0.03)$ | | | | | | | | | |
| 0.05 | 0.74 | 0.03 | 0.05 | 0.02 | 0.05 | 0.05 | 0.00 | 0.00 | 0.34 |
| 0.10 | 0.94 | 0.04 | 0.05 | 0.01 | 0.01 | 0.11 | 0.00 | 0.00 | 0.41 |
| 0.15 | 0.99 | 0.04 | 0.05 | 0.01 | 0.02 | 0.16 | 0.00 | 0.00 | 0.46 |
| 0.20 | 1.00 | 0.05 | 0.05 | 0.01 | 0.01 | 0.24 | 0.00 | 0.00 | 0.53 |
| 0.25 | 1.00 | 0.05 | 0.07 | 0.00 | 0.00 | 0.32 | 0.00 | 0.00 | 0.60 |
| 0.30 | 1.00 | 0.05 | 0.10 | 0.00 | 0.00 | 0.41 | 0.00 | 0.00 | 0.69 |
| **Easy, Solvable:** NE1 $= (0.01, 0.01)$, NE2 $= (0.07, 0.24)$ | | | | | | | | | |
| 0.05 | 0.81 | 0.03 | 0.02 | 0.02 | 0.04 | 0.04 | 0.00 | 0.01 | 0.34 |
| 0.10 | 0.99 | 0.02 | 0.02 | 0.01 | 0.02 | 0.09 | 0.00 | 0.00 | 0.40 |
| 0.15 | 1.00 | 0.02 | 0.01 | 0.00 | 0.00 | 0.17 | 0.00 | 0.00 | 0.46 |
| 0.20 | 1.00 | 0.02 | 0.01 | 0.00 | 0.00 | 0.24 | 0.00 | 0.00 | 0.54 |
| 0.25 | 1.00 | 0.02 | 0.01 | 0.00 | 0.00 | 0.32 | 0.00 | 0.00 | 0.61 |
| 0.30 | 1.00 | 0.02 | 0.01 | 0.00 | 0.00 | 0.41 | 0.00 | 0.00 | 0.67 |
| **Hard, Not Solvable:** NE1 $= (0.16, 1.00)$, NE2 $= (0.20, 0.88)$ | | | | | | | | | |
| 0.05 | 0.67 | 0.20 | 0.90 | 0.17 | 0.76 | 0.06 | 0.04 | 0.14 | 0.32 |
| 0.10 | 0.91 | 0.19 | 0.91 | 0.13 | 0.63 | 0.10 | 0.02 | 0.07 | 0.39 |
| 0.15 | 0.97 | 0.19 | 0.92 | 0.10 | 0.41 | 0.18 | 0.01 | 0.03 | 0.45 |
| 0.20 | 0.99 | 0.19 | 0.95 | 0.07 | 0.25 | 0.24 | 0.01 | 0.01 | 0.51 |
| 0.25 | 1.00 | 0.19 | 0.96 | 0.03 | 0.13 | 0.31 | 0.00 | 0.00 | 0.58 |
| 0.30 | 1.00 | 0.19 | 0.96 | 0.02 | 0.06 | 0.39 | 0.00 | 0.00 | 0.66 |
| **Easy, Not Solvable:** NE1 $= (0.09, 0.32)$, NE2 $= (0.14, 0.56)$ | | | | | | | | | |
| 0.05 | 0.68 | 0.13 | 0.51 | 0.10 | 0.37 | 0.05 | 0.02 | 0.07 | 0.33 |
| 0.10 | 0.97 | 0.12 | 0.53 | 0.08 | 0.28 | 0.10 | 0.01 | 0.05 | 0.39 |
| 0.15 | 1.00 | 0.12 | 0.52 | 0.05 | 0.17 | 0.16 | 0.01 | 0.03 | 0.46 |
| 0.20 | 1.00 | 0.12 | 0.53 | 0.03 | 0.08 | 0.23 | 0.01 | 0.03 | 0.52 |
| 0.25 | 1.00 | 0.12 | 0.48 | 0.02 | 0.05 | 0.31 | 0.00 | 0.02 | 0.59 |
| 0.30 | 1.00 | 0.12 | 0.48 | 0.01 | 0.04 | 0.39 | 0.00 | 0.01 | 0.65 |

Table 1: Summary of the outcome of the synthetic studies. Columns labeled WPP1 refer to results obtained for $\underline{\beta} = \bar{\beta} = 1$, while WPP2 refers to the case $\underline{\beta} = 0.9, \bar{\beta} = 1.1$. The first column is the level in which we set the remaining parameters, $\epsilon_x = \epsilon_y = \epsilon_w = k_\epsilon$. The second column is the frequency by which a WPP solution has been found among 100 runs. For each particular method (NE1, NE2, Faithfulness and WPP) we report the pair (error average, error tail mass at 0.1), as explained in the main text. The Faithfulness estimator is the back-door adjustment obtained by using as the admissible set the same set found by WPP1. Averages are taken only over the cases where a witness-admissible set pair has been found. The columns following each WPP results are the median width of the respective WPP interval across the 100 runs.

Manski (2007) emphasizes, this is the price for making fewer assumptions. Even there, they typically cover only about 25% of the interval $[-1, 1]$ of *a priori* possibilities for the ACE.

### 6.3 Influenza Study

Our empirical study concerns the effect of influenza vaccination on a patient being later on hospitalized with chest problems. $X = 1$ means the patient got a flu shot, $Y = 1$ indicates the patient was hospitalized. A negative ACE therefore suggests a desirable vaccine. The study was originally discussed by McDonald et al. (1992). Shots were not randomized, but doctors were randomly assigned to receive a reminder letter to encourage their patients to be inoculated, an event recorded as binary variable *GRP*. This suggests the standard IV model in Figure 1(d), with $W = GRP$ and $U$ unobservable. That is, $W$ and $U$ are independent because $W$ is randomized, and there are resonable justifications to believe the lack of a direct effect of letter randomization on patient hospitalization. Richardson et al. (2011) and Hirano et al. (2000) provide further discussion.

From this randomization, it is possible to directly estimate the ACE[16] of $W$ on $Y$: $-0.01$. This is called *intention-to-treat* (ITT) analysis (Rothman et al., 2008), as it is based on the treatment assigned by randomization and not on the variable of interest $(X)$, which is not randomized. While the ITT can be used for policy making, the ACE of $X$ on $Y$ would be a more interesting result, as it reveals features of the vaccine that are not dependent on the encouragement design. $X$ and $Y$ can be confounded, as $X$ is not controlled. For instance, the patient choice of going to be vaccinated might be caused by her general health status, which will be a factor for hospitalization in the future.

The data contains records of $2,681$ patients, with some demographic indicators (age, sex and race) and some historical medical data (for instance, whether the patient is diabetic). A total of 9 covariates is available. Using the bounds of Balke and Pearl (1997) and observed frequencies gives an interval of $[-0.23, 0.64]$ for the ACE. WPP could *not* validate *GRP* as a witness for any admissible set.

Instead, when forbidding *GRP* to be included in an admissible set (since the theory says *GRP* cannot be a common direct cause of vaccination and hospitalization), WPP selected as the highest-scoring pair the witness *DM* (patient had history of diabetes prior to vaccination) with admissible set composed of *AGE* (dichotomized as "60 or less years old," and "above 60") and *SEX*. Choosing, as an illustration, $\epsilon_w = \epsilon_y = \epsilon_x = 0.2$ and $\underline{\beta} = 0.9$, $\bar{\beta} = 1.1$, we obtain the posterior expected interval $[-0.10, 0.17]$. This does *not* mean the vaccine is more likely to be bad (positive ACE) than good: the posterior distribution is over bounds, not over points, being completely agnostic about the distribution within the bounds. Notice that even though we allow for full dependence between all of our variables, the bounds are stricter than in the standard IV model due to the weakening of hidden confounder effects postulated by observing conditional independences. It is also interesting that two demographic variables ended up being chosen by Rule 1, instead of other indicators of past diseases.

When allowing *GRP* to be included in an admissible set, the pair (*DM*, *AGE*, *SEX*) is now ranked second among all pairs that satify Rule 1, with the first place being given

---

16. Notice that while the ACE might be small, this does not mean that in another scale, such as odd-ratios, the results do not reveal an important effect. This depends on the domain.

Figure 5: Scatterplot of the joint posterior distribution of lower bounds and upper bounds, Pearson correlation coefficient of 0.71.

by *RENAL* as the witness (history of renal complications), with the admissible set being *GRP, COPD* (history of pulmonary disease), and *SEX*. In this case, the expected posterior interval was approximately the same, $[-0.07, 0.16]$. It is worthwhile to mention that, even though this pair scored highest by our criterion that measures the posterior probability distribution of each premise of Rule 1, it is clear that the fit of this model is not as good as the one with *DM* as the witness, as measured by the much larger proportion of rejected samples when generating the posterior distribution. This suggests future work on how to rank such models.

In Figure 5 we show a scatter plot of the posterior distribution over lower and upper bounds on the influenza vaccination, where *DM* is the witness. In Figure 6(a) and (b) we show kernel density estimators based on the Monte Carlo samples for the cases where *DM* and *RENAL* are the witnesses, respectively. While the witnesses were tested using the analytical bounds, the final set of samples shown here were generated with the fully numerical optimization procedure, which is quite expensive.

We also analyze how Algorithm ?? and its variants for $\tau_w$ and $\tau_c$ can be used to select $\aleph = \{\epsilon_w, \epsilon_x, \epsilon_y, \underline{\beta}, \bar{\beta}\}$. The motivation is that this is a domain with overall weak dependencies among variables. From one point of view, this is bad as instruments will be weak and generate wide intervals (as suggested by Proposition 1). From another perspective, this suggests that the effect of hidden confounders may also be weak.

Following the framework of Algorithm 3 in Section 5.2, we put independent uniform $[0, 1]$ priors on the relaxation parameters $\epsilon_w, \epsilon_x = \epsilon_y$ and $\underline{\beta} = 1/\bar{\beta}$. 8 admissible sets provide

Figure 6: In (a), the marginal densities for the lower bound (red) and upper bound (blue) on the ACE, smoothed kernel density estimates based on 5000 Monte Carlo samples. Bounds were derived using $DM$ as the witness. In (b), a similar plot using $RENAL$ as the witness.

the reference set for the target set $(DM, (AGE, SEX))$, where we disallow the empty set and any admissible set containing $GRP$. Reference set back-door adjusted ACEs are all very weak. Figure 7 shows the posterior inference and a Gaussian density estimate of the distribution of reference ACEs using the empirical distribution to estimate each individual ACE.

The result is used to define strongly informative relaxation parameters $0.56, 0.02$ and $0.99$, with a corresponding expected posterior interval of $[0.01, 0.02]$, suggesting a deleterious effect of the vaccination. The 95% posterior credible interval for the lower bound, however, also includes zero. While we do not claim by any means that this procedure provides irrefutable ACE bounds for this problem (such is the case for any observational study), this illustrates that, even for a small number of covariates, there is an opportunity to use reasonably broad priors and obtain informative consequences on the values of ℵ by a more conservative exploitation of the faithfulness assumption of Spirtes et al. (2000).

## 7. Conclusion

Our model provides a novel compromise between point estimators given by the faithfulness assumption and bounds based on instrumental variables. We believe such an approach

Figure 7: In (a), MCMC plots for the relaxation parameters $\epsilon_w$ (red curve), $\epsilon_x = \epsilon_y$ (blue curve) and $\underline{\beta} = 1/\bar{\beta}$ (black curve) using the framework of Section 5.2. The respective means are 0.38, 0.02 and 0.99. In (b), a Gaussian fit for the chosen ACEs used to generate the posterior over the relaxation parameters (mean $-0.0007$, standard deviation 0.0009).

should become a standard item in the toolbox of anyone who needs to perform an observational study[17].

Unlike risky Bayesian approaches that put priors directly on the parameters of the unidentifiable latent variable model $P(Y, X, W, U \mid \mathbf{Z})$, the constrained Dirichlet prior on the observed distribution does not suffer from massive sensitivity to the choice of hyperparameters. By focusing on bounds, WPP keeps inference more honest. While it is tempting to look for an alternative that will provide a point estimate of the ACE, it is also important to have a method that trades-off information for fewer assumptions. WPP provides a framework to express such assumptions.

As future work, we will look at a generalization of the procedure beyond relaxations of chain structures $W \to X \to Y$. Much of the machinery here developed, including Entner et al.'s Rules, can be adapted to the case where causal ordering is unknown: starting from the algorithm of Mani et al. (2006) to search for "Y-structures," it is possible to generalize Rule 1 to setups where we have an outcome variable $Y$ that needs to be controlled, but where there is no covariate $X$ known not to be a cause of other covariates. Finally, the techniques used to derive the symbolic bounds in Section 4 may prove useful in a more general context,

17. R code for all methods is available at http://www.homepages.ucl.ac.uk/∼ucgtrbd/wpp.

and complement other methods to find subsets of useful constraints such as the graphical approach of Evans (2012).

## Acknowledgments

We thank McDonald, Hiu and Tierney for their flu vaccine data. Most of this work was done while RS was hosted by the Department of Statistics at the University of Oxford.

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

## Appendix A. Proofs

In this Appendix, we prove the results mentioned in the main text.

**Proof of Proposition 1** In the standard IV case, simple analytical bounds are known for $P(Y = y \mid do(X = x))$ (Balke and Pearl, 1997; Dawid, 2003):

$$\eta_0 \leq \min \begin{cases} 1 - \zeta_{00.0} \\ 1 - \zeta_{00.1} \\ \zeta_{01.0} + \zeta_{10.0} + \zeta_{10.1} + \zeta_{11.1} \\ \zeta_{10.0} + \zeta_{11.0} + \zeta_{01.1} + \zeta_{10.1} \end{cases} \qquad \eta_0 \geq \max \begin{cases} \zeta_{10.1} \\ \zeta_{10.0} \\ \zeta_{10.0} + \zeta_{11.0} - \zeta_{00.1} - \zeta_{11.1} \\ -\zeta_{00.0} - \zeta_{11.0} + \zeta_{10.1} + \zeta_{11.1} \end{cases}$$

$$\eta_1 \leq \min \begin{cases} 1 - \zeta_{01.1} \\ 1 - \zeta_{01.0} \\ \zeta_{10.0} + \zeta_{11.0} + \zeta_{00.1} + \zeta_{11.1} \\ \zeta_{00.0} + \zeta_{11.0} + \zeta_{10.1} + \zeta_{11.1} \end{cases} \qquad \eta_1 \geq \max \begin{cases} \zeta_{11.1} \\ \zeta_{11.0} \\ -\zeta_{01.0} - \zeta_{10.0} + \zeta_{10.1} + \zeta_{11.1} \\ \zeta_{10.0} + \zeta_{11.0} - \zeta_{01.1} - \zeta_{10.1} \end{cases}$$

where $\eta_x \equiv P(Y = 1 \mid do(X = x))$ and $\zeta_{yx.w} \equiv P(Y = y, X = x \mid W = w)$. Define also $\alpha_x \equiv P(Y = 1 \mid X = x)$ and $\beta_w \equiv P(X = 1 \mid W = w)$ so that

$$\zeta_{yx.w} = \alpha_x^{I(y=1)} (1 - \alpha_x)^{I(y=0)} \beta_w^{I(x=1)} (1 - \beta_w)^{I(x=0)}, \tag{23}$$

where $I(\cdot)$ is the indicator function returning 1 or 0 depending on whether its argument is true or false, respectively.

Assume for now that $\beta_1 \geq \beta_0$, that is, $P(X = 1 \mid W = 1) \geq P(X = 1 \mid W = 0)$. We will first show that $1 - \zeta_{00.0} \leq \min\{1 - \zeta_{00.1}, \zeta_{01.0} + \zeta_{10.0} + \zeta_{10.1} + \zeta_{11.1}, \zeta_{10.0} + \zeta_{11.0} + \zeta_{01.1} + \zeta_{10.1}\}$.

That $1 - \zeta_{00.0} \leq 1 - \zeta_{00.1}$ follows directly from the relationship (23) and the assumptions $W \per\!\!\!\perp Y \mid X$ and $\beta_1 \geq \beta_0$: $(1 - \zeta_{00.0}) - (1 - \zeta_{00.1}) = -(1 - \alpha_0)(1 - \beta_0) + (1 - \alpha_0)(1 - \beta_1) = (1 - \alpha_0)(\beta_0 - \beta_1) \leq 0$.

Now consider $(1 - \zeta_{00.0}) - (\zeta_{01.0} + \zeta_{10.0} + \zeta_{10.1} + \zeta_{11.1})$. This is equal to

$$\begin{aligned} &= (1 - (1 - \alpha_0)(1 - \beta_0)) - ((1 - \alpha_1)\beta_0 + \alpha_0(1 - \beta_0) + \alpha_0(1 - \beta_1) + \alpha_1\beta_1) \\ &= (\beta_0 + \alpha_0(1 - \beta_0)) - (\beta_0 - \alpha_1\beta_0 + \alpha_0(1 - \beta_0) + \alpha_0 - \alpha_0\beta_1 + \alpha_1\beta_1) \\ &= \alpha_1(\beta_0 - \beta_1) - \alpha_0(1 - \beta_1) \leq 0 \end{aligned}$$

Analogously, we can show that $1 - \zeta_{00.0} \leq \zeta_{10.0} + \zeta_{11.0} - \zeta_{01.1} - \zeta_{10.1}$. Tedious but analogous manipulations lead to the overall conclusion

$$1 - \zeta_{00.0} = \min \begin{cases} 1 - \zeta_{00.0} \\ 1 - \zeta_{00.1} \\ \zeta_{01.0} + \zeta_{10.0} + \zeta_{10.1} + \zeta_{11.1} \\ \zeta_{10.0} + \zeta_{11.0} + \zeta_{01.1} + \zeta_{10.1} \end{cases} \qquad \zeta_{10.0} = \max \begin{cases} \zeta_{10.1} \\ \zeta_{10.0} \\ \zeta_{10.0} + \zeta_{11.0} - \zeta_{00.1} - \zeta_{11.1} \\ -\zeta_{00.0} - \zeta_{11.0} + \zeta_{10.1} + \zeta_{11.1} \end{cases}$$

$$1 - \zeta_{01.1} = \min \begin{cases} 1 - \zeta_{01.1} \\ 1 - \zeta_{01.0} \\ \zeta_{10.0} + \zeta_{11.0} + \zeta_{00.1} + \zeta_{11.1} \\ \zeta_{00.0} + \zeta_{11.0} + \zeta_{10.1} + \zeta_{11.1} \end{cases} \qquad \zeta_{11.1} = \max \begin{cases} \zeta_{11.1} \\ \zeta_{11.0} \\ -\zeta_{01.0} - \zeta_{10.0} + \zeta_{10.1} + \zeta_{11.1} \\ \zeta_{10.0} + \zeta_{11.0} - \zeta_{01.1} - \zeta_{10.1} \end{cases}$$

The upper bound on the ACE $\eta_1 - \eta_0$ is obtained by subtracting the lower bound on $\eta_0$ from the upper bound on $\eta_1$. That is, $\eta_1 - \eta_0 \leq (1 - \zeta_{01.1}) - \zeta_{10.0} = \mathcal{U}_{SIV}$. Similarly, $\eta_1 - \eta_0 \geq \zeta_{11.1} - (1 - \zeta_{00.0}) = \mathcal{L}_{SIV}$. It follows that $\mathcal{U}_{SIV} - \mathcal{L}_{SIV} = 1 - (P(X = 1 \mid W = 1) - P(X = 1 \mid W = 0))$.

Finally, assuming $\beta_1 \leq \beta_0$ gives by symmetry the interval width $1 - (P(X = 1 \mid W = 0) - P(X = 1 \mid W = 1))$, implying the width in the general case is given by $1 - |P(X = 1 \mid W = 1) - P(X = 1 \mid W = 0)|$. ∎

Now we will prove the main theorems stated in Section 4. To facilitate reading, we repeat here the notation used in the description of the constraints with a few additions, as well as the identities mapping different parameter spaces and the corresponding assumptions exploited in the derivation.

We start with the basic notation,

$$\begin{aligned} \zeta^{\star}_{yx.w} &\equiv P(Y = y, X = x \mid W = w, U) \\ \zeta_{yx.w} &\equiv \sum_U P(Y = y, X = x \mid W = w, U)P(U \mid W = w) \\ &= P(Y = y, X = x \mid W = w) \\ \kappa_{yx.w} &\equiv \sum_U P(Y = y, X = x \mid W = w, U)P(U) \end{aligned}$$

$$\begin{aligned} \eta^{\star}_{xw} &\equiv P(Y = 1 \mid X = x, W = w, U) \\ \eta_{xw} &\equiv \sum_U P(Y = 1 \mid X = x, W = w, U)P(U \mid W = w) \\ &= P(Y = 1 \mid do(X = x), W = w) \\ \omega_{xw} &\equiv \sum_U P(Y = 1 \mid X = x, W = w, U)P(U) \end{aligned}$$

$$\begin{aligned} \delta^{\star}_w &\equiv P(X = 1 \mid W = w, U) \\ \delta_w &\equiv \sum_U P(X = 1 \mid W = w, U)P(U \mid W) = P(X = 1 \mid W = w) \\ &= \zeta_{11.w} + \zeta_{01.w} \\ \chi_{x.w} &\equiv \sum_U P(X = x \mid W = w, U)P(U) \\ &= \kappa_{1x.w} + \kappa_{0x.w} \end{aligned}$$

The explicit relationship between parameters describing the latent variable model is:

$$
\begin{aligned}
\zeta^\star_{00.0} &= (1 - \eta^\star_{00})(1 - \delta^\star_0) \\
\zeta^\star_{01.0} &= (1 - \eta^\star_{10})\delta^\star_0 \\
\zeta^\star_{10.0} &= \eta^\star_{00}(1 - \delta^\star_0) \\
\zeta^\star_{11.0} &= \eta^\star_{10}\delta^\star_0 \\
\zeta^\star_{00.1} &= (1 - \eta^\star_{01})(1 - \delta^\star_1) \\
\zeta^\star_{01.1} &= (1 - \eta^\star_{11})\delta^\star_1 \\
\zeta^\star_{10.1} &= \eta^\star_{01}(1 - \delta^\star_1) \\
\zeta^\star_{11.1} &= \eta^\star_{11}\delta^\star_1
\end{aligned}
$$

All upper bound constants $U^{.U}_{..}$ are assumed to be positive. For $L^{.U}_{..} = 0$, $c \geq 0$, all ratios $c/L^{.U}_{..}$ are defined to be positive infinite.

In what follows, we define "the standard IV model" as the one which obeys exogeneity of $W$ and exclusion restriction – that is, the model following the directed acyclic graph $\{W \to X \to Y, X \leftarrow U \to Y\}$. All variables are binary, and the goal is to bound the average causal effect (ACE) of $X$ on $Y$ given a non-descendant $W$ and a possible (set of) confounder(s) $U$ of $X$ and $Y$.

**Proof of Theorem 2** Start with the relationship between $\eta_{xw}$ and its upper bound:

$$
\begin{aligned}
\eta^\star_{xw} &\leq U^{YU}_{xw} & \text{(Multiply both sides by } \delta^\star_{x'.w}) \\
\eta^\star_{xw}(1 - (1 - \delta^\star_{x'.w})) &\leq U^{YU}_{xw}\delta^\star_{x'.w} & \text{(Marginalize over } P(U)) \\
\omega_{xw} - \kappa_{1x.w} &\leq U^{YU}_{xw}\chi_{x'.w} \\
\omega_{xw} &\leq \kappa_{1x.w} + U^{YU}_{xw}(\kappa_{0x'.w} + \kappa_{1x'.w})
\end{aligned}
$$

and an analogous series of steps gives $\omega_{xw} \geq \kappa_{1x.w} + L^{YU}_{xw}(\kappa_{0x'.w} + \kappa_{1x'.w})$. Notice such bounds above will depend on how tight $\epsilon_y$ is. As an illustration of its implications, consider the derived identity $\zeta^\star_{0x.w} = (1 - \eta^\star_{xw})\delta^\star_{x.w} \Rightarrow 1 - \eta^\star_{xw} = \zeta^\star_{0x.w}/\delta^\star_{x.w} \Rightarrow 1 - \eta^\star_{xw} \geq \zeta^\star_{0x.w} \Rightarrow$ $\eta^\star_{xw} \leq 1 - \zeta^\star_{0x.w} = \zeta^\star_{0x.w} + \zeta^\star_{0x'.w} + \zeta^\star_{1x'.w} \Rightarrow \omega_{xw} \leq \kappa_{0x.w} + \kappa_{0x'.w} + \kappa_{1x'.w}$.

It follows from $U^{YU}_{xw} \leq 1$ that that the derived bound $\omega_{xw} \leq \kappa_{1x.w} + U^{YU}_{xw}(\kappa_{0x'.w} + \kappa_{1x'.w})$ is at least as tight as the one obtained via $\eta^\star_{xw} \leq 1 - \zeta^\star_{0x.w}$. Notice also that the standard IV bound $\eta_{xw} \leq 1 - \zeta_{0x.w}$ (Balke and Pearl, 1997; Dawid, 2003) is a special case for $\epsilon_y = 0$, $\underline{\beta} = \bar{\beta} = 1$.

For the next bounds, consider

$$
\begin{aligned}
\delta^\star_{x.w} &\leq U^{XU}_{xw} \\
\eta^\star_{xw}\delta^\star_{x.w} &\leq U^{XU}_{xw}\eta^\star_{xw} & \text{(Marginalize over } P(U)) \\
\kappa_{1x.w} &\leq U^{XU}_{xw}\omega_{xw} \\
\omega_{xw} &\geq \kappa_{1x.w}/U^{XU}_{xw}
\end{aligned}
$$

where the bound $\omega_{xw} \leq \kappa_{1x.w}/L^{XU}_{xw}$ can be obtained analogously. The corresponding bound for the standard IV model (with possible direct effect $W \to Y$) is $\eta_{xw} \geq \zeta_{1x.w}$, obtained again by choosing $\epsilon_x = 1$, $\underline{\beta} = \bar{\beta} = 1$. The corresponding bound $\omega_{xw} \geq \kappa_{1x.w}$ is a looser bound for $U^{XU}_{xw} < 1$. Notice that if $L^{XU}_{xw} = 0$, the upper bound is defined as infinite.

Finally, the last bounds are similar to the initial ones, but as a function of $\epsilon_x$ instead of $\epsilon_y$:

$$
\begin{array}{rcl}
\delta^\star_{x.w} & \leq & U^{XU}_{xw} \\
(1-\eta^\star_{xw})\delta^\star_{x.w} & \leq & U^{XU}_{xw}(1-\eta^\star_{xw}) \quad \text{(Marginalize over } P(U)) \\
\kappa_{0x.w} & \leq & U^{XU}_{xw}(1-\omega_{xw}) \\
\omega_{xw} & \leq & 1 - \kappa_{0x.w}/U^{XU}_{xw}
\end{array}
$$

The lower bound $\omega_{xw} \geq 1 - \kappa_{0x.w}/L^{XU}_{xw}$ is obtained analogously, and implied to be minus infinite if $L^{XU}_{xw} = 0$. ∎

**Proof of Theorem 3** We start with the following derivation,

$$
\begin{array}{rcl}
\eta^\star_{xw'} - \eta^\star_{xw} & \leq & \epsilon_w \\
\eta^\star_{xw'}\delta^\star_{x.w'} - \eta^\star_{xw}\delta^\star_{x.w'} & \leq & \epsilon_w\delta^\star_{x.w'} \quad \text{(Use } -U^{XU}_{xw'} \leq -\delta^\star_{x.w'}) \\
\eta^\star_{xw'}\delta^\star_{x.w'} - \eta^\star_{xw}U^{XU}_{xw'} & \leq & \epsilon_w\delta^\star_{x.w'} \quad \text{(Marginalize over } P(U)) \\
\kappa_{1x.w'} - \omega_{xw}U^{XI}_{xw} & \leq & \epsilon_w\chi_{x.w'} \\
\omega_{xw} & \geq & (\kappa_{1x.w'} - \epsilon_w\chi_{x.w'})/U^{XU}_{xw'} \\
\omega_{xw} & \geq & (\kappa_{1x.w'} - \epsilon_w(\kappa_{0x.w'} + \kappa_{1x.w'}))/U^{XU}_{xw'}
\end{array}
$$

Analogously, starting from $\eta^\star_{xw'} - \eta^\star_{xw} \geq \epsilon_w$, we obtain $\omega_{xw} \leq (\kappa_{1x.w'} + \epsilon_w(\kappa_{0x.w'} + \kappa_{1x.w'}))/L^{XU}_{xw'}$. Notice that for the special case $\epsilon_w$ and $U^{XU}_{xw'} = 1$, we obtain the corresponding lower bound $\omega_{xw} \geq \kappa_{1x.w'}$ that relates $\omega$ and $\kappa$ across different values of $W$.

The result corresponding to the upper bound $\eta_{xw} \leq 1 - \zeta_{0x.w'}$ can be obtained as follows:

$$
\begin{array}{rcl}
\eta^\star_{xw'} - \eta^\star_{xw} & \geq & -\epsilon_w \\
1 + \eta^\star_{xw'} - 1 - \eta^\star_{xw} & \geq & -\epsilon_w \\
(1-\eta^\star_{xw}) - (1-\eta^\star_{xw'}) & \geq & -\epsilon_w \\
(1-\eta^\star_{xw})\delta^\star_{x.w'} - (1-\eta^\star_{xw'})\delta^\star_{x.w'} & \geq & -\epsilon_w\delta^\star_{x.w'} \\
(1-\eta^\star_{xw})U^{XU}_{xw'} - (1-\eta^\star_{xw'})\delta^\star_{x.w'} & \geq & -\epsilon_w\delta^\star_{x.w'} \quad \text{(Marginalize over } P(U)) \\
(1-\omega_{xw})U^{XU}_{xw'} - \kappa_{0x.w'} & \geq & -\epsilon_w\chi_{x.w'} \\
\omega_{xw} & \leq & 1 - (\kappa_{0x.w'} - \epsilon_w(\kappa_{0x.w'} + \kappa_{1x.w'}))/U^{XU}_{xw'}
\end{array}
$$

with the corresponding lower bound (non-trivial for $L^{XU}_{xw'} > 0$) given by $\omega^\star_{xw} \geq 1 - (\kappa_{0x.w'} + \epsilon_w(\kappa_{0x.w'} + \kappa_{1x.w'}))/L^{XU}_{xw'}$.

The final block of relationships can be derived as follows:

$$
\begin{array}{rcl}
\eta^\star_{xw} - \eta^\star_{xw'} & \leq & \epsilon_w \\
\eta^\star_{xw}\delta^\star_{x'.w} - \eta^\star_{xw'}\delta^\star_{x'.w} & \leq & \epsilon_w\delta^\star_{x'.w} \\
\eta^\star_{xw}(1-(1-\delta^\star_{x'.w})) - \eta^\star_{xw'}\delta^\star_{x'.w} & \leq & \epsilon_w\delta^\star_{x'.w} \quad \text{(Use } -U^{XU}_{x'w} \leq -\delta^\star_{x'.w}) \\
\eta^\star_{xw} - \eta^\star_{xw}(1-\delta^\star_{x'.w}) - \eta^\star_{xw'}U^{XU}_{x'w} & \leq & \epsilon_w\delta^\star_{x'.w} \quad \text{(Marginalize over } P(U)) \\
\omega_{xw} - \kappa_{1x.w} - \omega_{xw'}U^{XU}_{x'w} & \leq & \epsilon_w\chi_{x'.w} \\
\omega_{xw} - \omega_{xw'}U^{XU}_{x'w} & \leq & \kappa_{1x.w} + \epsilon_w(\kappa_{0x'.w} + \kappa_{1x'.w})
\end{array}
$$

with the lower bound $\omega_{xw} - \omega_{xw'}L_{x'w}^{XU} \geq \kappa_{1x.w} - \epsilon_w(\kappa_{0x'.w} + \kappa_{1x'.w})$ derived analogously. Moreover,

$$
\begin{aligned}
\eta_{xw'}^{\star} - \eta_{xw}^{\star} &\leq \epsilon_w \\
(1 - \eta_{xw}^{\star})\delta_{x'.w}^{\star} - (1 - \eta_{xw'}^{\star})\delta_{x'.w}^{\star} &\leq \epsilon_w \delta_{x'.w}^{\star} \\
(1 - \eta_{xw}^{\star})(1 - (1 - \delta_{x'.w}^{\star})) - (1 - \eta_{xw'}^{\star})U_{x'w}^{XU} &\leq \epsilon_w \delta_{x'.w}^{\star} \\
1 - \omega_{xw} - \kappa_{0x.w} - (1 - \omega_{xw'})U_{x'w}^{XU} &\leq \epsilon_w \chi_{x'.w} \\
\omega_{xw} - \omega_{xw'}U_{x'w}^{XU} &\geq 1 - \kappa_{0x.w} - U_{x'w}^{XU} - \epsilon_w(\kappa_{0x'.w} + \kappa_{1x'.w})
\end{aligned}
$$

and the corresponding $\omega_{xw} - \omega_{xw'}L_{x'w}^{XU} \leq 1 - \kappa_{0x.w} - L_{x'w}^{XU} + \epsilon_w(\kappa_{0x'.w} + \kappa_{1x'.w})$. The last two relationships follow immediately from the definition of $\epsilon_w$. ∎

Our constraints found so far collapse to some of the constraints found in the standard IV models (Balke and Pearl, 1997; Dawid, 2003) given $\epsilon_w = 0$, $\underline{\beta} = \bar{\beta} = 1$. Namely,

$$
\begin{aligned}
\eta_{xw} &\leq 1 - \zeta_{0x.w} \\
\eta_{xw} &\leq 1 - \zeta_{0x.w'} \\
\eta_{xw} &\geq \zeta_{1x.w} \\
\eta_{xw} &\geq \zeta_{1x.w'}
\end{aligned}
$$

However, none of the constraints so far found counterparts in the following:

$$
\begin{aligned}
\eta_{xw} &\leq \zeta_{0x.w} + \zeta_{1x.w} + \zeta_{1x.w'} + \zeta_{1x'.w'} \\
\eta_{xw} &\leq \zeta_{0x.w'} + \zeta_{1x.w'} + \zeta_{1x.w} + \zeta_{1x'.w} \\
\eta_{xw} &\geq \zeta_{1x.w} + \zeta_{1x'.w} - \zeta_{0x.w'} - \zeta_{1x'.w'} \\
\eta_{xw} &\geq \zeta_{1x.w'} + \zeta_{1x'.w'} - \zeta_{0x.w} - \zeta_{1x'.w}
\end{aligned}
$$

These constraints have the distinct property of being functions of both $P(Y = x, X = x \mid W = w)$ and $P(Y = x, X = x \mid W = w')$, simultaneously. So far, we have only used the basic identities and constraints, without attempting at deriving constraints that are not a direct application of such identities. In the framework of (Dawid, 2003; Ramsahai, 2012), it is clear that general linear combinations of functions of $\{\delta_{x.w}^{\star}\eta_{1x.w}^{\star}, \delta_{x.w}^{\star}, \eta_{1x.w}^{\star}\}$ can generate constraints on observable quantities $\zeta_{yx.w}$ and causal quantities of interest, $\eta_{xw}$. We need to emcompass these possibilities in a way we get a framework for generating symbolic constraints as a function of $\{\epsilon_w, \epsilon_y, \epsilon_x, \underline{\beta}, \bar{\beta}\}$.

One of the difficulties on exploiting a black-box polytope package for that is due to the structure of the process, which exploits the constraints in Section 3 by first finding the extreme points of the feasible region of $\{\delta_w^{\star}\}$, $\{\eta_{xw}^{\star}\}$. If we use the constraints

$$
\begin{aligned}
|\eta_{x1}^{\star} - \eta_{x0'}^{\star}| &\leq \epsilon_w \\
0 \leq \eta_{xw}^{\star} &\leq 1
\end{aligned}
$$

then assuming $0 < \epsilon_w < 1$, we always obtain the following six extreme points

$$
\begin{aligned}
&(0, 0) \\
&(0, \epsilon_w) \\
&(\epsilon_w, 0) \\
&(1 - \epsilon_w, 1) \\
&(1, 1 - \epsilon_w) \\
&(1, 1)
\end{aligned}
$$

In general, however, once we introduce constraints $L_{xw}^{YU} \leq \eta_{xw}^\star \leq U_{xw}^{XU}$, the number of extreme points will vary. Moreover, when multiplied with the extreme points of the space $\delta_1^\star \times \delta_0^\star$, the resulting extreme points of $\zeta_{yx.w}^\star$ might be included or excluded of the polytope depending on the relationship among $\{\epsilon_w, \epsilon_x, \epsilon_y\}$ and the observable $P(Y, X \mid W)$. Numerically, this is not a problem (barring numerical instabilities, which do occur with a nontrivial frequency). Algebraically, this makes the problem considerably complicated[18]. Instead, in what follows we will define a simpler framework that will not give tight constraints, but will shed light on the relationship between constraints, observable probabilities and the $\epsilon$ parameters. This will also be useful to scale up the full Witness Protection Program, as discussed in the main paper.

## Methodology for Cross-W Constraints

Consider the standard IV model again, i.e., where $W$ is exogenous with no direct effect on $Y$. So far, we have not replicated anything such as e.g. $\eta_1 \leq \zeta_{00.0} + \zeta_{11.0} + \zeta_{10.1} + \zeta_{11.1}$. We call this a "cross-W" constraint, as it relates observables under different values of $W \in \{0, 1\}$. These are important when considering weakening the effect $W \to Y$. The recipe for deriving them will be as follows. Consider the template

$$\delta_0^\star f_1(\eta_0^\star, \eta_1^\star) + \delta_1^\star f_2(\eta_0^\star, \eta_1^\star) + f_3(\eta_0^\star, \eta_1^\star) \geq 0 \tag{24}$$

such that $f_i(\cdot, \cdot)$ are linear. Linearity is imposed so that this function will correspond to a linear function of $\{\zeta^\star, \eta^\star, \delta^\star\}$, of which expectations will give observed probabilities or interventional probabilities.

We will require that evaluating this expression at each of the four extreme points of the joint space $(\delta_0^\star, \delta_1^\star) \in \{0, 1\}^2$ will translate into one of the basic constraints $1 - \eta_i^\star \geq 0$ or $\eta_i^\star \geq 0$, $i \in \{0, 1\}$. This implies any combination of $\{\delta_0^\star, \delta_1^\star, \eta_0^\star, \eta_1^\star\}$ will satisfy (24) (more on that later).

Given a choice of basic constraint (say, $\eta_1^\star \geq 0$), and setting $\delta_0^\star = \delta_1^\star = 0$, this immediately identifies $f_3(\cdot, \cdot)$. We assign the constraint corresponding to $\delta_0^\star = \delta_1^\star = 1$ with the "complementary constraint" for $\eta_1$ (in this case, $\eta_1^\star \leq 1$). This leaves two choices for assigning the remaining constraints.

Why do we associate the $\delta_0^\star = \delta_1^\star = 1$ case with the complementary constraint? Let us parameterize each function as $f_i(\eta_0^\star, \eta_1^\star) \equiv a_i \eta_0^\star + b_i \eta_1^\star + c_i$. Let $a_3 = q$, where either $q = 1$ (case $\eta_0^\star \geq 0$) or $q = -1$ (case $1 - \eta_0^\star \geq 0$). Without loss of generality, assume case $(\delta_0^\star = 1, \delta_1^\star = 0)$ is associated with the complementary constraint where the coefficient of $\eta_0^\star$ should be $-q$. For the other two cases, the coefficient of $\eta_0^\star$ should be 0 by construction. We get the system

$$
\begin{aligned}
a_3 &= q \\
a_1 + a_3 &= -q \\
a_2 + a_3 &= 0 \\
a_1 + a_2 + a_3 &= 0
\end{aligned}
$$

---

18. As a counterpart, imagine we defined a polytope through the matrix inequality $A\mathbf{x} \leq \mathbf{b}$. If we want to obtain its extreme point representation as an algebraic function of the entries of matrix $A$ and vector $\mathbf{b}$, this will be a complicated problem since we cannot assume we know the magnitudes and signs of the entries.

This system has no solution. Assume instead $\delta_0^\star = \delta_1^\star = 1$ is associated with the complementary constraint where the coefficient of $\eta_0^\star$ should be $-q$. The system now is:

$$
\begin{aligned}
a_3 &= q \\
a_1 + a_3 &= 0 \\
a_2 + a_3 &= 0 \\
a_1 + a_2 + a_3 &= -q
\end{aligned}
$$

This system always have the solution $a_1 = a_2 = -q$. We do have freedom with $b_1, b_2, b_3$, which means we can choose to allocate the remaining two cases in two different ways.

**Lemma 6** *Consider the constraints derived by the above procedure. Then any choice of $(\delta_0^\star, \delta_1^\star, \eta_0^\star, \eta_1^\star) \in [0,1]^4$ will satisfy these constraints.*

**Proof** Without loss of generality, let $f_3(\eta_0^\star, \eta_1^\star) = q\eta_0^\star + (1-q)/2$, $q \in \{-1, 1\}$. That is, $a_3 = q, b_3 = 0, c_3 = (1-q)/2$. This implies $a_1 = a_2 = -q$ (as above). Associating $(\delta_0^\star = 1, \delta_1^\star = 0)$ with $\eta_1^\star \geq 0$ gives $\{b_1 = 1, c_1 = (q-1)/2\}$ and consequently associating $(\delta_0^\star = 0, \delta_0^\star = 1)$ with $1 - \eta_1^\star \geq 0$ implies $\{b_2 = -1, c_2 = (1+q)/2\}$. Plugging this into the expression $\delta_0^\star f_1(\eta_0^\star, \eta_1^\star) + \delta_1^\star f_2(\eta_0^\star, \eta_1^\star) + f_3(\eta_0^\star, \eta_1^\star)$ we get

$$
\begin{aligned}
&= \delta_0^\star(-q\eta_0^\star + \eta_1^\star + (q-1)/2) + \delta_1^\star(-q\eta_0^\star - \eta_1^\star + (1+q)/2) + q\eta_0^\star + (1-q)/2 \\
&= \eta_0^\star(q - (\delta_0^\star + \delta_1^\star)q) + \eta_1^\star(\delta_0^\star - \delta_1^\star) + \delta_0^\star(q-1)/2 + \delta_1^\star(1+q)/2 + (1-q)/2 \\
&= \eta_0^\star(q - (\delta_0^\star + \delta_1^\star)q) + \eta_1^\star(\delta_0^\star - \delta_1^\star) + (-q + (\delta_0^\star + \delta_1^\star)q)/2 + (\delta_1^\star - \delta_0^\star + 1)/2 \\
\\
&= q((\delta_1^\star + \delta_0^\star) - 1)(1 - 2\eta_0^\star)/2 + ((\delta_1^\star - \delta_0^\star)(1 - 2\eta_1^\star) + 1)/2 \\
&= (\delta_1^\star + \delta_0^\star - 1)s/2 + (\delta_1^\star - \delta_0^\star)t/2 + 1/2
\end{aligned}
$$

where $s = q(1 - 2\eta_0^\star) \in [-1,1]$ and $t = (1 - 2\eta_1^\star) \in [-1,1]$. Then evaluating at the four extreme points $s, t \in \{-1, +1\}$ we get $\delta_0, \delta_1, 1 - \delta_0, 1 - \delta_1$, all of which are non-negative. ∎

The procedure derives 8 bounds (4 cases that we get by associating $f_3$ with either $\eta_x \geq 0$ or $1 - \eta_x \geq 0$. For each of these cases, 2 subcases what we get by assigning $(\delta_0^\star = 1, \delta_1^\star = 0)$ with either $\eta_{x'} \geq 0$ or $1 - \eta_{x'} \geq 0$). Now, for an illustration of one case:

**Deriving a constraint for the standard IV model, example:** $f_3(\eta_0^\star, \eta_1^\star) \equiv \eta_0^\star \geq 0$

Associate $\eta_1^\star \geq 0$ with assigment $(\delta_0^\star = 1, \delta_1^\star = 0)$ (implying we associate $\eta_1^\star \leq 1$ with assigment $(\delta_0^\star = 0, \delta_1^\star = 1)$ and $\eta_0^\star \leq 1$ with $(\delta_0^\star = 1, \delta_1^\star = 1)$). This uniquely gives $f_1(\eta_0^\star, \eta_1^\star) = \eta_1^\star - \eta_0^\star$, $f_2(\eta_0^\star, \eta_1^\star) = -\eta_1^\star - \eta_0^\star + 1$. The resulting expression is

$$
\delta_0^\star(\eta_1^\star - \eta_0^\star) + \delta_1^\star(-\eta_1^\star - \eta_0^\star + 1) + \eta_0^\star \geq 0
$$

from which we can verify that the assignment $(\delta_0^\star = 1, \delta_1^\star = 1)$ gives $\eta_0^\star \leq 1$. Now, we need to take the expectation of the above with respect to $U$ to obtain observables $\zeta$ and causal distributions $\eta$. However, first we need some rearrangement so that we match $\eta_0^\star$ with corresponding $(1 - \delta_w^\star)$ and so on.

$$
\begin{aligned}
\eta_1^\star(\delta_0^\star - \delta_1^\star) + \eta_0^\star(1 - \delta_0^\star - \delta_1^\star) + \delta_1^\star &\geq 0 \\
\eta_1^\star(\delta_0^\star - \delta_1^\star) + \eta_0^\star((1 - \delta_0^\star) + (1 - \delta_1^\star) - 1) + \delta_1^\star &\geq 0 \\
\zeta_{11.0}^\star - \zeta_{11.1}^\star + \zeta_{10.0}^\star + \zeta_{10.1}^\star - \eta_0^\star + \zeta_{01.1}^\star + \zeta_{11.1}^\star &\geq 0
\end{aligned}
$$

Taking expectations and rearranging it, we have

$$\eta_0 \leq \zeta_{11.0} + \zeta_{10.0} + \zeta_{10.1} + \zeta_{01.1}$$

rediscovering one of the IV bounds for $\eta_0$. Choosing to associate $\eta_1^\star \geq 0$ with assigment $(\delta_0^\star = 0, \delta_1^\star = 1)$ will give instead

$$\eta_0 \leq \zeta_{11.1} + \zeta_{10.1} + \zeta_{10.0} + \zeta_{01.0}$$

Basically the effect of one of the two choices within any case is to switch $\zeta_{yx.w}$ with $\zeta_{yx.w'}$. ∎

### Deriving Cross-W Constraints

What is left is a generalization of that under the condition $|\eta_{xw} - \eta_{xw'}| \leq \epsilon_w$, $w \neq w'$, instead of $\eta_{xw} = \eta_{xw'}$. In this situation, we exploit the constraint $\underline{L} \leq \eta_{xw}^\star \leq \bar{U}$ instead of $0 \leq \eta_{xw}^\star \leq 1$ or $L_{xw}^{YU} \leq \eta_{xw}^\star \leq U_{xw}^{YU}$, where $\underline{L} \equiv \min\{L_{xw}^{YU}\}, \bar{U} \equiv \max\{U_{xw}^{YU}\}$. Using $L_{xw}^{YU} \leq \eta_{xw}^\star \leq U_{xw}^{YU}$ complicates things considerably. Also, we will not derive here the analogue proof of Lemma 1 for the case where $(\eta_0^\star, \eta_1^\star) \in [\underline{L}, \bar{U}]^2$, as it is analogous but with a more complicated notation.

**Proof of Theorem 4** We demonstrate this through two special cases.
General Model, Special Case 1: $f_3(\eta_{0w}^\star, \eta_{1w}^\star) \equiv \eta_{xw}^\star - \underline{L} \geq 0$

There are two modifications. First, we perform the same associations as before, but with respect to $\underline{L} \leq \eta_{xw}^\star \leq \bar{U}$ instead of $0 \leq \eta_x^\star \leq 1$. Second, before we take expectations, we swap some of the $\eta_{xw}^\star$ with $\eta_{xw'}^\star$ up to some error $\epsilon_w$.

Following the same sequence as in the example for the IV model, we get the resulting expression (where $x' \equiv \{0, 1\} \backslash x$):

$$\delta_w^\star(\eta_{x'w}^\star - \eta_{xw}^\star) + \delta_{w'}^\star(-\eta_{x'w}^\star - \eta_{xw}^\star + \bar{U} + \underline{L}) + \eta_{xw}^\star - \underline{L} \geq 0$$

from which we can verify that the assignment $(\delta_w^\star = 1, \delta_{w'}^\star = 1)$ gives $\bar{U} - \eta_{xw}^\star \geq 0$. Now, we need to take the expectation of the above with respect to $U$ to obtain "observables" $\kappa$ and causal effects $\omega$. However, the difficulty now is that terms $\eta_{xw}^\star \delta_{w'}^\star$ and $\eta_{xw'}^\star \delta_w^\star$ have no observable counterpart under expectation. We get around this transforming $\eta_{xw'}^\star \delta_w^\star$ into $\eta_{xw}^\star \delta_w^\star$ (and $\eta_{xw}^\star \delta_{w'}^\star$ into $\eta_{xw'}^\star \delta_{w'}^\star$) by adding the corresponding correction $-\eta_{xw}^\star \leq -\eta_{xw'}^\star + \epsilon_w$:

$$
\begin{aligned}
\delta_w^\star(\eta_{x'w}^\star - \eta_{xw}^\star) + \delta_{w'}^\star(-\eta_{x'w}^\star - \eta_{xw}^\star + \bar{U} + \underline{L}) + \eta_{xw}^\star - \underline{L} &\geq 0 \\
\delta_w^\star(\eta_{x'w}^\star - \eta_{xw}^\star) + \delta_{w'}^\star(-\eta_{x'w'}^\star + \epsilon_w - \eta_{xw'}^\star + \epsilon_w + \bar{U} + \underline{L}) + \eta_{xw}^\star - \underline{L} &\geq 0 \\
\eta_{x'w}^\star \delta_w^\star + \eta_{xw}^\star(1 - \delta_w^\star) - \eta_{x'w'}^\star \delta_{w'}^\star - \eta_{xw'}^\star \delta_{w'}^\star + \delta_{w'}^\star(\bar{U} + \underline{L} + 2\epsilon_w) - \underline{L} &\geq 0
\end{aligned}
$$

Now, the case for $x = 1$ gives

$$
\begin{aligned}
\eta_{0w}^\star \delta_w^\star + \eta_{1w}^\star(1 - \delta_w^\star) - \eta_{0w'}^\star \delta_{w'}^\star - \eta_{1w'}^\star \delta_{w'}^\star + \ldots &\geq 0 \\
\eta_{0w}^\star(1 - (1 - \delta_w^\star)) + \eta_{1w}^\star(1 - \delta_w^\star) - \eta_{0w'}^\star(1 - (1 - \delta_{w'}^\star)) - \eta_{1w'}^\star \delta_{w'}^\star + \ldots &\geq 0
\end{aligned}
$$

Taking the expectations:

$$\omega_{0w} - \kappa_{10.w} + \omega_{1w} - \kappa_{11.w} - \omega_{0w'} + \kappa_{10.w'} - \kappa_{11.w'} + \chi_{w'}(\bar{U} + \underline{L} + 2\epsilon_w) - \underline{L} \geq 0 \qquad (25)$$

Notice that for $\underline{\beta} = \bar{\beta} = 1$, $\underline{L} = 0$, $\bar{U} = 1$, $\epsilon_w = 0$, this implies $\eta_{xw} = \eta_{xw'}$ and this collapses to

$$\eta_{0w} - \zeta_{10.w} + \eta_{1w} - \zeta_{11.w} - \eta_{0w'} + \zeta_{10.w'} - \zeta_{11.w'} + \delta_{w'} \geq 0$$

$$\eta_{1w} \geq \zeta_{10.w} + \zeta_{11.w} - \zeta_{10.w'} - \zeta_{01.w'}$$

which is one of the lower bounds one obtains under the standard IV model.

The case for $x = 0$ is analogous and gives

$$\omega_{0w'} \leq \kappa_{11.w} + \kappa_{10.w} + \kappa_{10.w'} - \kappa_{11.w'} + \chi_{w'}(\bar{U} + \underline{L} + 2\epsilon_w) - \underline{L} \qquad (26)$$

The next subcase is when we exchange the assignment of $(\delta_w^\star, \delta_{w'}^\star)$ to other constraints. We obtain the following inequality:

$$\delta_{w'}^\star(\eta_{x'w}^\star - \eta_{xw}^\star) + \delta_w^\star(-\eta_{x'w}^\star - \eta_{xw}^\star + \bar{U} + \underline{L}) + \eta_{xw}^\star - \underline{L} \geq 0$$

which from an analogous sequence of steps leads to

$$\delta_{w'}^\star(\eta_{x'w}^\star - \eta_{xw}^\star) + \delta_w^\star(-\eta_{x'w}^\star - \eta_{xw}^\star + \bar{U} + \underline{L}) + \eta_{xw}^\star - \underline{L} \geq 0$$
$$\delta_{w'}^\star(\eta_{x'w'}^\star + \epsilon_w - \eta_{xw'}^\star + \epsilon_w) + \delta_w^\star(-\eta_{x'w}^\star - \eta_{xw}^\star + \bar{U} + \underline{L}) + \eta_{xw}^\star - \underline{L} \geq 0$$
$$\eta_{x'w'}^\star\delta_{w'}^\star - \eta_{xw'}^\star\delta_{w'}^\star + 2\delta_{w'}^\star\epsilon_w - \eta_{x'w}^\star\delta_w^\star + \eta_{xw}^\star(1 - \delta_w^\star) + \delta_w^\star(\bar{U} + \underline{L}) - \underline{L} \geq 0$$

For $x = 1$,

$$\eta_{0w'}^\star\delta_{w'}^\star - \eta_{1w'}^\star\delta_{w'}^\star + \eta_{0w}^\star\delta_w^\star + \eta_{1w}^\star(1 - \delta_w^\star) + \ldots \geq 0$$
$$\eta_{0w'}^\star(1 - (1 - \delta_{w'}^\star)) - \eta_{1w'}^\star\delta_{w'}^\star - \eta_{0w}^\star(1 - (1 - \delta_w^\star)) + \eta_{1w}^\star(1 - \delta_w^\star) + \ldots \geq 0$$

Taking expectations,

$$\omega_{0w'} - \kappa_{10.w'} - \kappa_{11.w'} - \omega_{0w} + \kappa_{10.w} + \omega_{1w} - \kappa_{11.w} + 2\chi_{w'}\epsilon_w + \chi_w(\bar{U} + \underline{L}) - \underline{L} \geq 0 \qquad (27)$$

For $x = 0$,

$$\eta_{1w'}^\star\delta_{w'}^\star - \eta_{0w'}^\star\delta_{w'}^\star + \eta_{1w}^\star\delta_w^\star + \eta_{0w}^\star(1 - \delta_w^\star) + \ldots \geq 0$$
$$\eta_{1w'}^\star\delta_{w'}^\star - \eta_{0w'}^\star(1 - (1 - \delta_{w'}^\star)) - \eta_{1w}^\star\delta_w^\star + \eta_{0w}^\star(1 - \delta_w^\star) + \ldots \geq 0$$
$$\kappa_{11.w'} - \omega_{0w'} + \kappa_{10.w'} - \kappa_{11.w} + \kappa_{10.w} + 2\chi_{w'}\epsilon_w + \chi_w(\bar{U} + \underline{L}) - \underline{L} \geq 0$$

$$\omega_{0w'} \leq \kappa_{11.w'} + \kappa_{10.w'} - \kappa_{11.w} + \kappa_{10.w} + 2\chi_{w'}\epsilon_w + \chi_w(\bar{U} + \underline{L}) - \underline{L} \qquad (28)$$

General Model, Special Case 2: $f_3(\eta_{0w}^\star, \eta_{1w}^\star) \equiv \bar{U} - \eta_{xw}^\star \geq 0$

Associate $\eta_{x'w}^\star \geq \underline{L}$ with assigment $(\delta_w^\star = 1, \delta_{w'}^\star = 0)$ (implying we associate $\eta_{x'w}^\star \leq \bar{U}$ with assigment $(\delta_w^\star = 0, \delta_{w'}^\star = 1)$ and $\eta_{xw}^\star \geq \underline{L}$ with $(\delta_w^\star = 1, \delta_{w'}^\star = 1)$). The resulting expression is

$$\delta_w^\star(\eta_{x'w}^\star + \eta_{xw}^\star - \bar{U} - \underline{L}) + \delta_{w'}^\star(-\eta_{x'w}^\star + \eta_{xw}^\star) + \bar{U} - \eta_{xw}^\star \geq 0$$

Following the same line of reasoning as before, we get this for $x = 1$:

$$\omega_{0w} - \omega_{0w'} - \omega_{1w} - \kappa_{10.w} + \kappa_{11.w} + \kappa_{10.w'} + \kappa_{11.w'} - \chi_w(\bar{U} + \underline{L}) + 2\epsilon_w\chi_{w'} + \bar{U} \geq 0 \quad (29)$$

We get this for $x = 0$:

$$\omega_{0w'} \geq -\kappa_{11.w} + \kappa_{10.w} + \kappa_{11.w'} + \kappa_{10.w'} + \chi_w(\bar{U} + \underline{L}) - 2\epsilon_w\chi_{w'} - \bar{U} \quad (30)$$

With the complementary assignment, we start with the relationship

$$\delta_{w'}^{\star}(\eta_{x'w}^{\star} + \eta_{xw}^{\star} - \bar{U} - \underline{L}) + \delta_w^{\star}(-\eta_{x'w}^{\star} + \eta_{xw}^{\star}) + \bar{U} - \eta_{xw}^{\star} \geq 0$$

For $x = 1$,

$$\omega_{0w'} - \omega_{0w} - \omega_{1w} - \kappa_{10.w'} + \kappa_{11.w'} + \kappa_{10.w} + \kappa_{11.w} + \chi_{w'}(2\epsilon_w - \bar{U} - \underline{L}) + \bar{U} \geq 0 \quad (31)$$

For $x = 0$,

$$\omega_{0w'} \geq -\kappa_{11.w'} + \kappa_{10.w'} + \kappa_{11.w} + \kappa_{10.w} - \chi_{w'}(2\epsilon_w - \bar{U} - \underline{L}) - \bar{U} \quad (32)$$

Notice that the bounds obtained are asymmetric in $x$, i.e., we derive different bounds for $\omega_{0w}$ and $\omega_{1w}$. Symmetry is readily obtained by the same derivation where $\delta_w^{\star}$ is interpreted as $P(X = 0 \mid W = w, U)$ and $x$ is swapped with $x'$. ∎