[Reviews · NeurIPS 2014]

Submitted by Assigned_Reviewer_15

This paper is concerned with the problem of estimating the average causal effect (ACE) between two (binary) variables when the causal structure over the covariates is not fully given. It considers a rule proposed in Entner et al. (2013) that aims to select a set of observed covariates that is sufficient to block all back-door paths. For Entner et al.’s purpose, however, the rule is justified only under the faithfulness assumption. This paper proposes to use the rule to provide bounds rather than point estimates of the ACE, and develops a method following Ramsahai (2012).

I have to admit that I haven’t digested all the technical details, but the main idea makes sense to me and seems interesting and useful. What puzzled me, however, was that I didn’t really see the role of the conditions in Entner et al.’s rule in this approach. Without the faithfulness assumption, those conditions do not have obvious implications for the causal structure. My best guess is that the conditions, if verified, are relevant to the choice of the relaxation parameters, but the paper explicitly puts that matter aside. It seems to me that the machinery does not crucially depend on the conditions in Entner et al.’s rule, and I wonder if the apparent robustness of the method in the experiments has anything to do with this. I would appreciate a clarification on this issue.

Some minor comments:

1. First paragraph of Section 3, “… express violations of faithfulness as bounded violations of local independence …” This sounds strange. Isn’t unfaithfulness a matter of allowing extra independence rather than violating independence? It seems to me that the proposed framework is actually intended to allow/express “almost violations” of faithfulness by constraining the magnitude of dependence.

2. In the experiments, it may be helpful to also report the performance of Entner et al.’s method when their stringent thresholds are used, taking the answer of “don’t know” as reporting uninformative bounds.

3. In the second paragraph on p. 8, if “X=1 means the patient got a flu shot, Y=1 indicates the patient was *not* hospitalized”, why does a negative ACE suggest a desirable vaccine?

4. “Table 1” on top of p. 8 is cited as “Table 5” in the text.
Summary: This is an interesting paper, though the details are not easy to follow.

Submitted by Assigned_Reviewer_36

This paper proposes a new approach to derive the bound of the average causal effect (ACE) from a variable X to another Y. Its main principles is the use of Entner's Rule 1 in its inverse direction, the relaxation of the faithfulness assumption to introduce possibility of the path-cancellation at some degree, and the introduction of assumptions represented by some inequality constraints. A computation scheme of the ACE bound are formulated by using linear programing under these principles. Though the authors have not optimized the algorithm to derive the bounds yet, the resulted bounds seem very narrow and accurate in comparison with the other approaches in state of the art.

Quality
The authors tried to establish a novel method to derive more accurate bound of ACE than the state of the art. They combined several feasible principles and succeeded to provide the tighter bounds. In this regard, the content of the paper is very interesting.
However, some criteria to define the bounds are not comprehensively explained. Particularly, the validity of the use of Entner's Rule 1 in the inverse direction is not clearly discussed. For its use in the inverse direction, the authors should explicitly assume the structure depicted in Fig.2. This should be clearly explained and discussed.
In addition, the generic criteria to choose appropriate parameters for relaxation are not provided. Since these affect the result of the bounds significantly, the principle to determine the parameters should be more investigated and included in the work.

Clarity
The explanation of the paper on the theoretical part seems basically comprehensive except the essential discussion I pointed out above.
However, the demonstration of the experimental results are not well described. As I pointed out above, the actual implementation of the algorithm is not comprehensively explained. The definition of the parameters shown in Table 1 (Table 5?) are not clearly given.
In addition, I see some typos. For example;
"The full algorithm is shown in Table 1." on page 6=> I do not see any table of algorithm.
"Results are summarized in Table 5." on page 8=> I do not see Table 5 (maybe Table 1).

Originality
The problem setting of this paper has been addressed by some earlier work as the authors pointed out. In this regard, the scope of this paper does not have very string originality.
The idea to combine the three ideas in a unified shape contains some originality. But, as I pointed out above, the validity of the combination is not very clearly discussed.

Significance
In terms of deriving the tighter bounds of ACE, this method shows some significance. However, its performance has been check by quite limited experiments. It generality in both theory and wide experiments has not been strictly assessed.
Summary: The paper addresses an important problem which is to evaluate the bound of the strength of causal relations. However, the validity of the presented idea should be more well described in both theory and experiments.

Submitted by Assigned_Reviewer_42

The paper describes the so-called Witness Protection Program (WPP) algorithm to compute bounds on an inferred average causal effect (ACE) that takes possible violations of the standard faithfulness assumption in causal inference into account.
It does so by introducing a set of relaxation parameters to constrain unfaithful interactions, and then uses a linear programming framework to solve for the resulting upper and lower bounds on the ACE for a given link by adjusting for an appropriate set of covariates, subject to these constraints.

This problem currently receives a fair amount of attention, so the paper is both timely and interesting. I like the approach and ideas introduced in the paper: it is a principled attempt to explicitly quantify the impact of (weak) unfaithful interaction patterns, in combination with well-known mathematical tools to solve the corresponding maximization/minimization problems.

Whether or not they found the right method to capture this impact remains to be seen. The resulting output is hard to interpret from a conceptual point of view: a distribution over bounds which are a function of 'unfaithful relaxations' relative to a given model ... but how likely these unfaithful instances are, and with it how relevant/realistic these bounds themselves are remains unclear. Many relaxations lead to wide bounds to the point where they can become less than useful. The method does not commit to any preferred point estimate within the interval ... even though people are likely to interpret it that way.

Experimental evaluation is poor: the 'bias' evaluation metric is meaningless in comparing the WPP method with its competitors as they literally measure two completely different things. As a result it is unclear if the WPP output is a valuable addition to practitioners or merely a complicated distraction. Also the practical performance on the influenza data set is worryingly weak: ACE in (-0.23,0.64) implies next to no possible relevant conclusion ... although that could be the fairest assessment of them all.

Quality
The approach introduced in the paper is technically sound, with many details explicated in the supplementary file (I did not check any details in there). Ideas are well introduced, although the overall context is lost on occasion. Important aspects like choosing the relaxation parameters are relegated to a future article, which makes it difficult to get a feel for potential issues with this part of the method. Experimental evaluation is not balanced or robust enough.

Clarity
Decently written (with exception of conclusion, see comments below). The algorithmic steps and subsequent modifications in terms of many subtly different parameters was quite hard to follow, in part due to lack of explanation of certain details of the method. (I think I understand what happens, but I am not sure I could fully implement it from just this description; supplement may just be sufficient).

Originality
As far as I can tell the method is a novel combination of familiar techniques. It is closely related to, but sufficiently different from recent work on the same problem. As such it may stimulate a fruitful cross-fertilization of several approaches on the subject.

Significance
The significance of the paper mainly lies in the ideas and direction it brings to the problem, showing how to incorporate other techniques in solving this highly challenging problem. In its current implementation it seems unlikely to represent the finished article for practitioners, although some of the ideas employed in the WPP algorithm could well end up playing an important role.

Summary:
Relevant and highly topical problem. Interesting method that will undoubtedly find an audience. Current application limited, but likely to extend to other, more general cases. Reasonably well written, though could be improved. Technically interesting solutions. Experimental evaluation is poor. Relevance and proper interpretation of output distribution over bounds remains unclear.

----------------
Other comments:
p1/5: 'Witness Protection Program' - > great name for an algorithm
p1/47: 'unstable' is not the right word (they are stable, but may not hold true)
p2/74: 'cartoon representation'?
p2/93-94: technically this also assumes causal sufficiency or knowledge W - > X
p3/127: explain that witness W is itself not needed in the computation of the ACE, only as a means to identify set Z
p3/143: typo 'an useful'
p3/144: 'gives greater control over violations of faithfulness' : too vague as I do not see the control aspect
p4/164: might - > can;
idem: '(conditional)' - > remove or change to '(unconditional)' , as always W dep. U | X
p4/172-180: very sudden transition in level of abstraction; hard to interpret for readers less familiar with the subject - > explain / give some intuition on what these parameters represent,
p4/183: link to original introduction in (1),
p4/185-189: again interpret the constraint parameters; (easily misread as weak interactions)
p5/221: briefly describe the relation between the search part and the bounds computation,
idem: explain how the identification of admissible sets Z for W depends on the relaxation parameters theta
p5/239: 'Background ... used here.' - > sentence/purpose unclear: used how? important for the paper?
p5/259: to be clear: you mean max. 10 free parameters to describe the entire set Z, not a set of 10 variables, right?

p6/304: 'We know of no previous analytical bounds' - > This seems closely related to recent papers such as 'Estimating Causal Eff ects by Bounding Confounding' (Geiger, Janzing, Sch olkopf, UAI2014)
p6/305: as a last-minute reviewer I have not checked the supplementary material in detail
p6/319-332; I like the method but I doubt the practical efficacy of the back-substitution process in refining bounds; seems rare to be able to exploit such constraint violations, (can you give some idea of how effective this step is in section 5?)
p7/350: 'more than one bound' - > this is due to different possible admissible sets for different witnesses right?
p7/357: 'comparison is not straightforward' - > no: the subsequent trivial optimal example shows that comparison based on the bias metric is meaningless.
p7/360: 'only about 30% of the interval (-1,1)' - > that still covers (0,0.6) which implies 'anywhere between a strong causal link and no causal link' ...
p7/369: '5000 points' - > typical (medical) trials often have a lot less data available: worried about the impact on the size of the bounds for say 500 records,
p8/384: it is completely impossible to gauge meaningful information from Table 1 on the benefit of using WPP over the standard point estimate using faithfulness.
p8/391: 'WPP is quite stable' - > seems like a nice way of saying that the bounds derived by WPP are so loose that they don't even depend on whether the problem is solvable or not
p8/406-8: 'This does not mean ... within the bounds.' - > this is a crucial statement for interpretation: many casual readers will read the bounds as a confidence interval around the best guess value. It also highlights the difficulty in using the WPP results in practice: what do the bounds we derive actually mean?

p8/421+: 'risky Bayesian approaches', 'WPP bounds keep inference more honest' - > emotive language that suggests a biased evaluation and/or erroneous interpretation of results
idem: 'providing a compromise' - > WPP is not a compromise but focusses on an additional aspect of the overall inference problem,
idem: 'and purely theory-driven analyses that refuse to look at competing models' - > I have no idea what you refer to in this statement, but it definitely sounds unduly dismissive
p9: typo ref 9: inuenza
Summary: Relevant and highly topical problem. Technically and conceptually
interesting method that is likely to extend to more general cases.
Reasonably well written, though could be improved. Experimental
evaluation is poor. Relevance and proper interpretation of output
distribution over bounds remains unclear.
Author Feedback
Author rebuttal: We thank all reviewers for the feedback. We address points as space allows.

We do not agree that our definition of bias is "meaningless". The definition is the same as applied to all methods: the distance from the truth to the closest point in the given estimate, *whether the estimate is a single point or a set*. There is a trade-off: width of the interval against bias, width zero being just a special case. We can set parameters so that the width of WPP intervals is arbitrarily small, and so arbitrarily close to the faithfulness estimator. In the paper we use three widths: zero (==faithfulness), 0.3 (Table 1), 0.6 (SuppMat), and we discuss the trade-offs for a universal measure of bias. Apart from that, what we can do is to provide more combinations of priors, epsilon/beta at different widths etc. but essential conclusions remain the same: width of zero gives what we think is bad bias, we obtain small (near-zero) bias if one pays the price of an interval of width 0.3. Notice that starting from a backdoor-adjusted point estimate, there is no rationale on which way to grow an interval. WPP provides a principled way to set a trade-off with explicit assumptions. We know of no previous mass comparison of faithfulness vs bounding methods in the literature, even though it is a fundamental problem.

The above also suggests a way to choose parameters: set each epsilon to some k <= 1, beta_lower=1/c, beta_upper=c, c >= 1, tweaking so that the resulting interval width is something the analyst is willing to pay for. We can think of other default rules, *all of which are more flexible than faithfulness, and the loss of information quantifiable*. Unfortunately we cannot provide a thorough discussion of more sophisticated methods in the space that NIPS allows: e.g. [1] is a long paper itself, and our paper is already dense.

Influenza: [-0.23,0.64] is the IV interval [2], not ours. [9] has point estimates, strongly depends on prior regardless of sample size[18]; tighter bounds are only obtained by making more assumptions. These problems are not easy to resolve: Manski provides many reasons why bound analysis is relevant (25,000 citations in his Google Scholar page).

Our experimental setup is simple but informative, and shows that WPP can give robust answers that adequately account for the bias. Standard approaches (priors on the latent variable model) are not straightforward because of unidentifiability [18]. We were careful not to pick an experiment which was contrived to work for our method, instead it is a difficult set up with confounders and 'rogue' selection variables, including cases where there is no correct witness. We agree that more comprehensive studies are always a positive extension, and straight suggestions given editorial constraints are welcome: as it is, Assigned_Reviewers (AR) 36/42 propose no alternative evaluation method (we could have implemented one for the response).

Interpretation of distribution over bounds: this is standard Bayesian inference. Bounds are functions of unknown P(W,X,Y,Z) of which we only observe data: bounds need to be estimated, and estimators have uncertainty. One can ignore it [2], or find confidence [17] or posterior intervals [18] *over the endpoints of the bound*. E.g. what if I want a lower-bound for the ACE by smoothing P(W,X,Y,Z) conditioned on the constraints? Report its posterior expected value. But what if I want to account for uncertainty? Report e.g. the 0.05 quantile of the marginal lower bound posterior (shown in SuppMat for readers to make their own conclusions).

AR36: bounds probabilistic or deterministic? Constraints are deterministic *given* P(W,X,Y,Z) (line 194,248). But P(W,X,Y,Z) is unknown. Models are falsifiable, as in hypothesis testing: constraints (null hypothesis) are imposed on the joint, but the data might not agree with them. Line 5 could be a nested Bayesian model selection (saturated model vs constrained), hard to compute. We much prefer the computationally efficient rule of thumb of [18], which rejects a model when rejection sampling is slow.

AR36: validity of Rule 1 in context; AR42: p3/144; AR15: role of Rule 1. This is a direct application of faithfulness. Sketch: path W -> Y is ruled out, Z blocks all backdoors, so no W -- U -> {X, Y} possible either. Allowing any further dependence will be a relaxation of that. ACE bound optimization problem can be defined conditioned on the relaxation. With our parameterization (lines 171-179) the resulting problem generalizes [6]. That is all. E.g. paths W -- U -> Y, W -> Y, W -> X -> Y, may (statistically) cancel out. However, motivated by (in)dependencies involving Z and X, we assume they do not cancel out arbitrarily: the conditional dependence of Y and W given everybody else(crucially including U) is regulated by eps_W and assumed not to vary arbitrarily. In fact, *all eps/beta parameters regulate the degree of conditional dependence among UWXY* and at several points we discuss their interpretation at different values. This is unlike arbitrary faithfulness violations (where eps_W == 1) or faithfulness holding up (eps_W == 0). It is a relaxation of faithfulness by allowing (near) path cancellations, but not *any* (near) path cancellations.

AR36: parameters shown in Table 1. See 358-360
p1/47: unstable in the statistical sense
p2/93-94: no, not necessary, as there is no do(W) in the parameterization. See also [16]
p5/259: the dimension of Z refers to the number of binary variables.
p6/304: there are decades-old analytical bounds [11], but not of the optimized IV nature with extra parameters. In fact, our contribution is more than a combination of existing methods, as analytical bounds needed a technique different from [6,16].

We take seriously the point that the method is not that straightforward to implement (familiarity with e.g. [16] should suffice). Documented R code will be released. Thanks again for the detailed reviews, and many good suggestions.